neuroscience, behaviour

comprehensions, prosody, gesture, mouth, surprisal, N400

**Author for correspondence:**
Ye Zhang
e-mail: y.zhang.16@ucl.ac.uk

# More than words: word predictability, prosody, gesture and mouth movements in natural language comprehension

Ye Zhang[1], Diego Frassinelli[3], Jyrki Tuomainen[2], Jeremy I. Skipper[1] and Gabriella Vigliocco[1]

[1]Experimental Psychology, and [2]Experimental Psychology, Speech, Hearing and Phonetic Sciences, University College London, London, UK
[3]Department of Linguistics, University of Konstanz, Konstanz, Germany

YZ, 0000-0003-4697-783X; GV, 0000-0002-7190-3659

The ecology of human language is face-to-face interaction, comprising cues such as prosody, co-speech gestures and mouth movements. Yet, the multimodal context is usually stripped away in experiments as dominant paradigms focus on linguistic processing only. In two studies we presented video-clips of an actress producing naturalistic passages to participants while recording their electroencephalogram. We quantified multimodal cues (prosody, gestures, mouth movements) and measured their effect on a well-established electroencephalographic marker of processing load in comprehension (N400). We found that brain responses to words were affected by informativeness of co-occurring multimodal cues, indicating that comprehension relies on linguistic and non-linguistic cues. Moreover, they were affected by interactions between the multimodal cues, indicating that the impact of each cue dynamically changes based on the informativeness of other cues. Thus, results show that multimodal cues are integral to comprehension, hence, our theories must move beyond the limited focus on speech and linguistic processing.

## 1. Introduction

Language originated [1,2], is learnt [3–5] and is often used [6–8] in face-to-face contexts where comprehension takes advantage of both audition and vision. In face-to-face contexts, linguistic information is accompanied by multimodal 'non-linguistic' cues like speech intonation (prosody), hand gestures and mouth movements. Behavioural, neuroimaging and electrophysiological research has shown that these cues individually improve speech perception and comprehension in studies where other cues are carefully controlled or absent [6–9]. However, in the real world, these cues co-occur and we do not know whether what has been established in the laboratory isolating each cue holds in naturalistic contexts.

In studies that investigate individual multimodal cues, often, one cue is carefully manipulated while all the other possible interacting cues are either eliminated or kept constant. For example, prosody is normalized and auditory (rather than audiovisual) presentation is used when studying speech [10]; only the mouth, rather than the whole body is shown when studying audiovisual speech perception [11]; and the face is hidden when studying gestures [12]. Note that the focus on one modality is not only typical of communication research among humans. The situation is similar in research investigating animal communication. Here studies also typically focus on vocalization, or facial expressions or gestures (see reviews in [1,13]; however see [14,15]). This introduces two main problems. First, the materials and tasks often do not reflect real-world interactions: it is simply impossible not to see a person's gestures or mouth movements while they speak. Second, it breaks the natural and possibly

predictive correlation among cues with unknown consequences on processing [16,17]. The disruption of the relative reliability of cues can affect whether and how much the brain can rely on a given cue [18,19].

Increasing evidence suggests that comprehension involves generating predictions about upcoming units (sounds or words) based on prior linguistic context [20–22]. Prediction matters in language because it can constrain the interpretation of ambiguous sounds, words or sentences. A handful of previous studies have shown that multimodal (individual) cues can modulate such predictions [23]. However, we do not know whether non-linguistic audiovisual cues will impact on linguistic prediction during natural language comprehension.

In particular, two key questions need to be answered to develop theories that account for natural language comprehension. First, we need to understand to what extent the processing of multimodal cues is central in natural language processing (e.g. whether a cue is only used when the linguistic information is ambiguous, or in experimental tasks that force attention to it). Answering this question is necessary to properly frame theories because, if some multimodal cues (e.g. gesture or prosody) *always* contribute to processing, this would imply that our current focus mainly on linguistic information is too narrow, if not misleading. Second, we need to understand the dynamics of online multimodal comprehension. To provide mechanistic accounts of language comprehension, it is necessary to establish how the weight of a certain cue dynamically changes depending upon the presence and informativeness of other cues (e.g. whether the informativeness of mouth movements may play a greater or lesser role when meaningful gestures are also available).

## (a) The impact of prosody, gesture and mouth movements on processing: the state of the art

Accentuation (i.e. prosodic stress characterized as higher pitch making words acoustically prominent) marks new information [24]. Many behavioural studies have shown a comprehension benefit when new information is accentuated and old information is deaccentuated (indexed by faster response time) [25,26] and a cost when, instead there is incongruence between newness of information and accentuation. Such incongruence has been shown to correlate with increased activation in the left inferior frontal gyrus, suggesting increased processing difficulty [27]. In electrophysiological (EEG) studies, this mismatch elicits a more negative N400 than appropriate accentuation [28,29]. The N400 is an event-related-potential (ERP) peaking negatively around 400 ms after word presentation around central–parietal areas [30], which has been argued to index cognitive load and prediction in language comprehension [9].

Meaningful co-speech gestures (i.e. *iconic gestures* that imagistically evoke aspects of the meaning expressed, e.g. 'drinking'—shaping hands as if holding a cup and moving towards mouth; *emblematic gestures* that have conventionalized meanings, e.g. 'two'—holding out two fingers; *concrete deictic gestures* that points towards the entity being talked about, e.g. 'hair'—pointing towards the speakers' hair) improve comprehension by providing additional semantic information [31]. EEG studies showed that activating the less predictable meaning of homonymous words (e.g. 'dancing' for the word 'ball') using a gesture reduces the N400

response to a later mention of 'dance' [12]. Incongruence between meaningful gestures and linguistic context triggers more negative N400 compared with congruent gestures, indicating that gestures constrain predictions for upcoming words based on previous linguistic context [32,33]. Meaningful gestures activate posterior middle-temporal and inferior frontal regions, which have been interpreted in terms of the importance of these nodes for integration of speech and gestural information [34,35], although other studies have demonstrated separability between nodes involved in speech and gesture processing [36,37]. Moreover, the presence of meaningful gestures results in a significant reduction in cortical activity in auditory language regions (namely posterior superior temporal regions), a hallmark of prediction [38].

Fewer studies investigated beat gestures (i.e. meaningless gestures time-locked to the speech rhythm) [39]. Some argued that beats enhance the saliency of speech, similar to prosodic accentuation [40], and activate auditory cortex-like prosody [41]. Some studies found that beat gestures improved learning along with prosodic accentuation, (measured by better recall performances), but it remains controversial whether these two cues interact [42,43]. Two studies reported that presence of beat gestures induce less negative N400, similar to prosodic accentuation [43,44]. Other EEG studies, however, reported that beat gestures modulated brain responses in a later window (around 600 ms, interpreted as effects of beats on syntactic parsing [45] or on the integration of contextual information [46]).

Finally, many previous studies focused on the sensory-motor mechanism underscoring the use of mouth movements in speech, indicating that they facilitate the perception of auditory signals [47], modulate early sensory ERPs (N1–P2 reduction, associated with audiovisual integration) [48] and activate auditory cortices [49]. Less is known about whether mouth movements affect word predictability. Some behavioural and fMRI work suggests that mouth movements facilitate lexical access [50,51] and meaning comprehension [52] even when meaningful gestures are present [23,53]. However, while Brunellière and colleagues compared N400 of words starting with more or less informative mouth movements (/b/ v.s. /k/) and found that words with more informative mouth movements elicited more negative N400 [11], suggesting increased processing difficulty, Hernández-Gutiérrez and colleagues failed to find any N400 effect associated with mouth movements when comparing videos with dynamic facial and mouth movement and with a still image of the speaker [54].

Thus, previous studies indicate that at least when taken individually, multimodal cues interact with linguistic information in modulating the predictability of upcoming words. However, because studies only consider each non-linguistic cue individually, the natural and possibly predictive correlation among cues is altered with unknown consequences on processing [16,17]. Further, whether and how multimodal cues interact has not been fully explored. For example, the interaction between prosody and meaningful gestures has never been addressed, despite the fact that they are correlated [55].

## (b) The present study

We carried out two experiments (an original study and a replication with different materials) using materials that preserve the natural correlation across cues. Using a design that preserves ecological validity (figure 1), we address two key

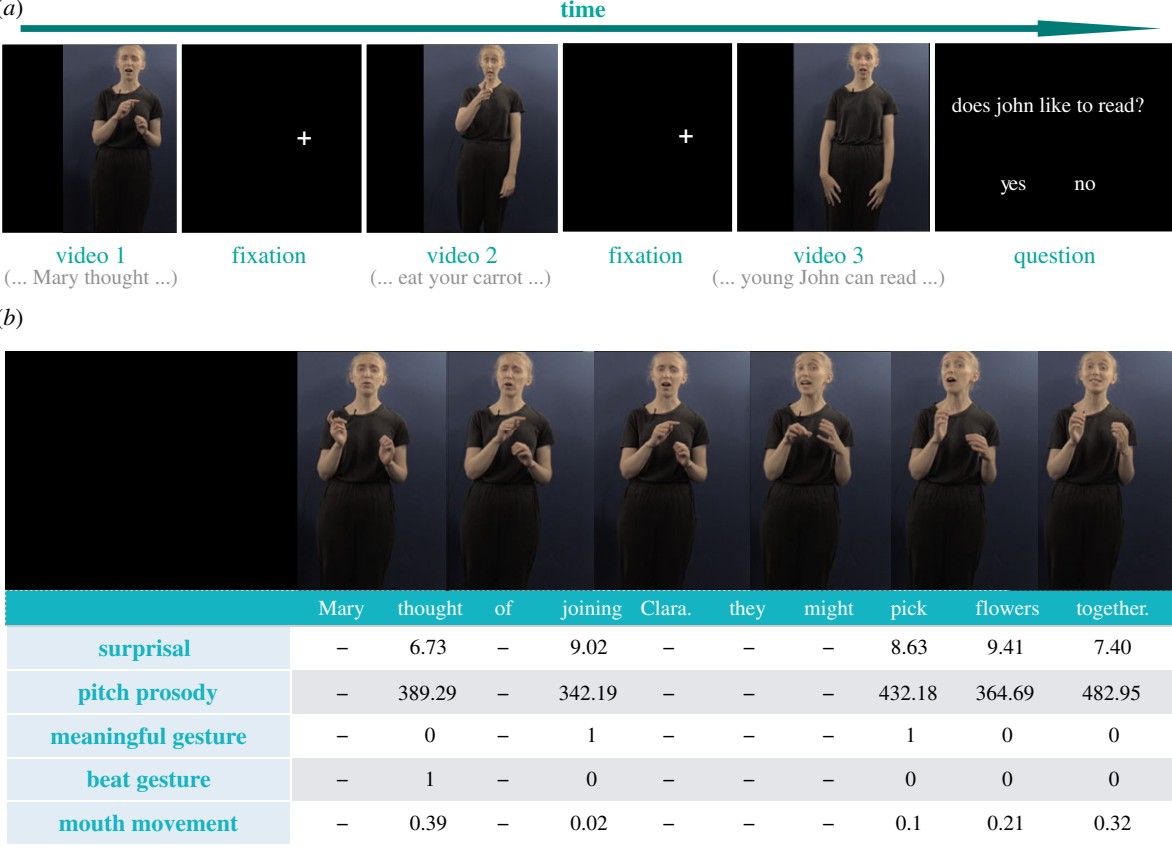

**Figure 1.** Illustration of study design. (*a*) Participants watched videos of an actress narrating short passages in naturalistic style, and answered comprehension questions following some videos. (*b*) We quantified informativeness of surprisal, pitch prosody, gestures and mouth movements per each content word. (Online version in colour.)

|  | Mary | thought | of | joining | Clara. | they | might | pick | flowers | together. |
|---|---|---|---|---|---|---|---|---|---|---|
| **surprisal** | – | 6.73 | – | 9.02 | – | – | – | 8.63 | 9.41 | 7.40 |
| **pitch prosody** | – | 389.29 | – | 342.19 | – | – | – | 432.18 | 364.69 | 482.95 |
| **meaningful gesture** | – | 0 | – | 1 | – | – | – | 1 | 0 | 0 |
| **beat gesture** | – | 1 | – | 0 | – | – | – | 0 | 0 | 0 |
| **mouth movement** | – | 0.39 | – | 0.02 | – | – | – | 0.1 | 0.21 | 0.32 |

questions about face-to-face multimodal communication. First, to what extent is the processing of multimodal (non-linguistic) cues central to natural language processing? We answer this question assessing whether the presence of any multimodal cue (and their combination) modulates predictions—based on prior discourse—for upcoming words indexed by N400. On the basis of past results, we predict that N400 amplitude will generally decrease for words with low predictability when any of the cues are present and informative. Second, what are the dynamics of online multimodal comprehension? We address this question by analysing the interaction between multimodal cues. If the weight of a certain cue dynamically changes depending upon the context, then its impact on word predictability should change as a function of other cues.

## 2. Material and methods

### (a) Participants

Native English speakers with normal hearing and normal/corrected to normal vision were paid £7.50 per hour to participate after giving written consent (experiment 1: *n* = 36, 5 excluded for technical issues; experiment 2: *n* = 20). All methods were approved by the local ethics committee.

### (b) Material

For experiment 1, 103 naturalistic passages were extracted from the British National Corpus [56]. We edited the first sentence to resolve ambiguities. The passages were unrelated to one another. In experiment 2, to better approximate real-life spoken language use, we chose 83 spoken passages from BBC TV scripts (see electronic supplementary material for passage selection criteria in experiments 1 and 2 and the list of the passages used).

A native British English-speaking actress produced the passages with natural speed, prosody and facial expressions. Thus, although not fully naturalistic, our materials preserve the natural co-occurrences among the different cues. We recorded two versions of each passage: one in which she was instructed to gesture freely and one in which she was instructed not to gesture. In the analyses, we compare the same word across with/without gesture conditions. In contrast to other cues such as prosody that are present for each word, gestures are not always produced and words likely to be accompanied by meaningful gestures (e.g. combing) are semantically very different from words that are not (e.g. pleasing) and these differences, unrelated to surprisal differences, could nonetheless be confounded. Thus, comparison of the same words produces clearer results. The actress has given informed consent for publication of identifying information.

### (c) Quantification of cues

The onset and offset of each word were automatically detected using a word-phoneme aligner based on a hidden Markov model [57] and was checked manually (word duration experiment 1: mean = 440 ms, s.d. = 376 ms; experiment 2: mean = 508 ms, s.d. = 306 ms). For each content word (i.e. nouns, adjectives, verbs and adverbs) we quantified the informativeness of each cue (linguistic predictability, pitch prosody, gesture and mouth). Function words (i.e. articles, pronouns, auxiliary verbs and prepositions) were excluded because Frank and colleagues failed to show any effect of the predictability (measured as surprisal) for such words [22].

*Linguistic predictability* was measured using surprisal (experiment 1: mean = 7.92, s.d. = 2.10; experiment 2: mean = 8.17, s.d. = 1.92), defined as the negative log-transformed conditional probability of a word given its preceding context [58]. Surprisal provides a good measure of predictability and predicts reading times [59] and N400 amplitude [22]. Here, surprisal was generated

using a bigram language model trained on the lemmatized version of the first slice (approx. 31-million tokens) of the ENCOW14-AX corpus [60]. Once trained, the model was used to calculate the surprisal of each word based on previous content words as follows:

$$\text{surprisal}(w_{t+1}) = -\log P(w_{t+1}|w_{1...t}),$$

where $w_{t+1}$ indicates the current word, and $w_{1...t}$ stands for previous content words (see electronic supplementary material for more details).

*Pitch prosody* per word was quantified as mean $F_0$ (experiment 1: mean = 298 Hz, s.d. = 84 Hz; experiment 2: mean = 288 Hz, s.d. = 88 Hz) extracted using Praat (v. 6.0.29) [61] (see electronic supplementary material for more details).

Gestures were coded as meaningful gestures or beats by expert coders (two in experiment 1, three in experiment 2) in ELAN (version 5.0.0) [62]. Meaningful gestures (experiment 1: $n = 359$; experiment 2: $n = 458$) comprised iconic gestures (e.g. drawing movements for the word 'drawing') and deictic gestures (e.g. pointing to the hair for 'hair'). Beat gestures (experiment 1: $n = 229$; experiment 2: $n = 340$) comprised rhythmic movements of the hands without clear meaning [39]. To associate words with gestures, two variables, meaningful gesture and beat gesture, were then created. Words received 1 for meaningful gesture if a corresponding meaningful gesture was annotated, and 1 for beat gesture if it overlapped with the stroke of a beat gesture.

*Mouth informativeness* (experiment 1: mean = 0.65, s.d. = 0.28; experiment 2: mean = 0.67, s.d. = 0.29) was quantified per word in online experiments (using Gorilla). An actress produced each word individually. Participants (recruited from Prolific) were paid £6 per hour to watch each word twice and guess the words based on the mouth shape. Every word was rated by 10 participants. We then calculated the reversed averaged phonological distance between the guesses and the answer using the Python library PanPhon [63]: larger value indicates more accurate guess thus higher mouth informativeness (see full details in electronic supplementary material).

## (d) Procedure
Participants watched videos (experiment 1: $n = 100$; experiment 2: $n = 79$), presented using Presentation software (v. 18.0), counterbalanced for gestures presence, while their EEG responses were recorded using a 32-channel BioSimi system (see full details in electronic supplementary material). Videos were separated by a 2000 ms interval in experiment 1, and 1000 ms interval in experiment 2. Some videos were followed by comprehension questions (Yes/No) about the content of the speech to ensure participants paid attention (experiment 1: 35 questions; experiment 2: 40 questions. See examples of questions and behavioural analysis in electronic supplementary material). They were instructed to watch videos carefully and answer the questions, when they were presented, as quickly and accurately as possible (prioritizing accuracy) by pressing the left (Yes) or right (No) control key. Participants sat approximately 1 m away from the screen (resolution = 1024 × 768) with 50Ω headphones. They were asked to avoid moving, keep their facial muscles relaxed and reduce blinking (when comfortable). The recording took ~30 min in experiment 1, ~60 min in experiment 2.

## (e) EEG analysis
Raw data were pre-processed with EEGLAB (v. 14.1.1) [64] and ERPLAB (v. 7.0.0) [65] in MATLAB (R2017b). See full details in electronic supplementary material. We first establish the time window where processing is affected by linguistic predictability in experiment 1. No previous study investigated surprisal effects in audiovisual communication. Therefore, rather than making *a priori* assumptions about the specific event-related response

we should observe, we carried out a hierarchical LInear MOdeling (LIMO toolbox [66]) to identify the EEG component sensitive to surprisal. This regression-based ERP analysis linearly decomposes the ERP signal into time-series of beta coefficient waveforms elicited by continuous variables (see full details in electronic supplementary material).

## (f) Linear mixed effect regression analysis
For both experiments, we performed linear mixed effect analysis (LMER) on the mean amplitude in 300–600 ms time window (determined by the results of LIMO analysis) using the R package lme4 [67]. We used LMER for its advantage in accommodating both categorical and continuous variables, thus increasing statistical power [68]. We excluded from the analyses: (a) words without a surprisal value (experiment 1: $n = 9$; experiment 2: $n = 13$); (b) words without a pitch prosody score (experiment 1: $n = 4$; experiment 2: $n = 2$); (c) words associated with both beat and meaningful gestures (experiment 1: $n = 3$; experiment 2: $n = 6$); (d) words occurring without any gesture in the 'with gesture' condition, and the corresponding words in without gesture videos (experiment 1: $n = 406$; experiment 2: $n = 685$, to avoid data unbalance). Mean ERP in the 300–600 ms and −100–0 ms time windows were extracted from 32 electrodes for each word as the dependent variable and the baseline. Owing to the likely overlap between baseline and the EEG signal of the previous word, we did not perform baseline correction during data pre-processing, but instead extracted the mean EEG amplitude in baseline interval and later included it in the regression model as control [22,69]. Independent variables included (1) predictors: surprisal, pitch prosody, meaningful gestures, beat gestures, mouth informativeness and all up to three-way interactions between surprisal and cues, excluding any meaningful*beat gestures interactions (instances where the two gestures co-occur were removed), (2) control: baseline, word length, word order in the sentence, sentence order in experiment and relative electrode positions measured by X, Y and Z coordinates [70]. Surprisal was log-transformed to normalize the data. All continuous variables were scaled so that coefficients represent the effect size. All categorical variables were sum coded so that coefficients represent the difference with the grand mean (intercept) [68]. We further included word lemma and participant as random variables. The maximal random structure failed to converge, so we included the highest interaction (three-way interactions) as random slope for participants [71], and surprisal as random slope for lemma (for experiment 1 only, the model failed to converge with surprisal as random slope in experiment 2). No predictors showed multicollinearity (experiment 1: VIF less than 2, kappa = 4.871; experiment 2: VIF less than 2.5, kappa = 5.76). Analysis of experiment 1 included 31 participants, 381 lemmas and 480 212 data points. Analysis of experiment 2 included 20 participants, 510 word-type lemmas and 434 944 data points. See full details of analysis in electronic supplementary material.

## 3. Results

## (a) Time window sensitive to linguistic context
Words with higher surprisal elicited more negative EEG response in the 300–600 ms time window especially in central–parietal areas. No other time window was significantly sensitive to surprisal (see plots in electronic supplementary material). As a result, we focused on the 300–600 ms time window in our subsequent analyses in both studies (analysis of experiment 2 yielded approximately the same window; see electronic supplementary material).

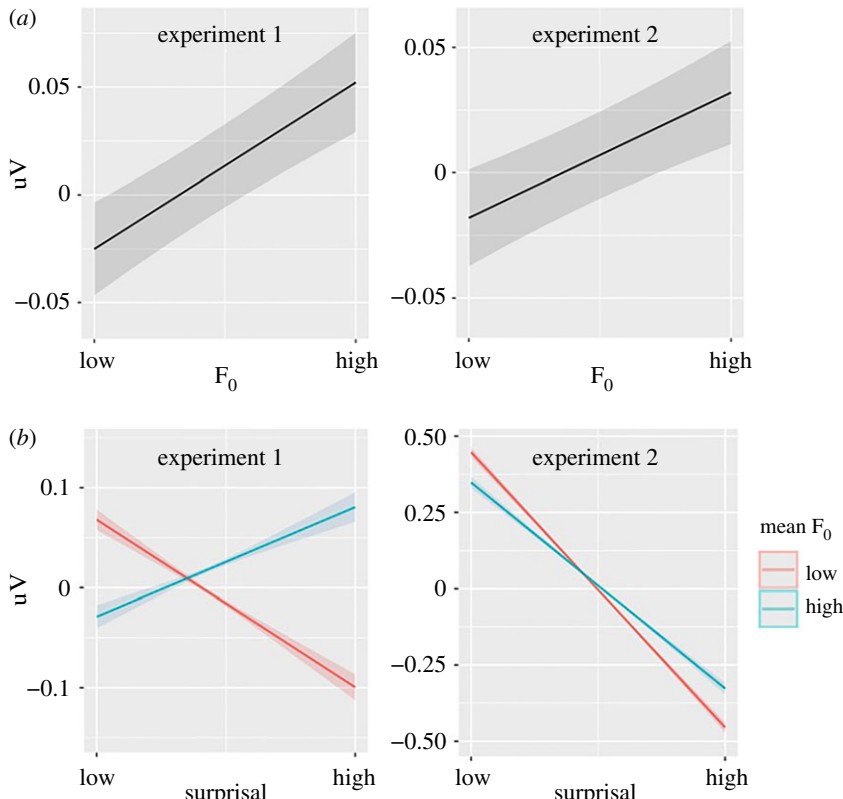

**Figure 2.** Prosodic accentuation (mean $F_0$) modulation of N400 amplitude. (*a*) Main effect of prosodic accentuation. (*b*) Interaction between prosodic accentuation and surprisal. Plots depict the predicted value of the mean amplitude of the ERP within 300–600 ms (grey areas = confidence intervals). All following conventions are the same. (Online version in colour.)

## (b) Are multimodal cues central to language processing?

To assess this question, we first focus on the main effects of the multimodal cues and their interaction with surprisal as predictors of N400 amplitude. Only replicable results are reported here and below (see full results in electronic supplementary material).

We found a main effect of pitch prosody (mean $F_0$, figure 2; experiment 1: $\beta = 0.011$, $p < 0.001$; experiment 2: $\beta = 0.014$, $p < 0.001$). Here and below, because all variables were standardised, a unit change in the IV corresponds to a unit change in $\beta$, thus representing effect sizes, although the values may not be directly comparable with Cohen's D. Words with higher pitch prosody showed less negative EEG, or smaller N400 amplitude. The interaction between surprisal and pitch prosody (experiment 1: $\beta = 0.022$, $p < 0.001$; experiment 2: $\beta = 0.012$, $p < 0.001$, accounting for 12% N400 change) indicates that pitch prosody modulates the N400 response associated with surprisal: high surprisal words showed a larger reduction of N400 amplitude when the pitch prosody was higher, in comparison to low surprisal words.

Meaningful gestures showed similar effects (figure 3). Words accompanied by a meaningful gesture showed a significantly less negative N400 (experiment 1: $\beta = 0.006$, $p < 0.001$; experiment 2: $\beta = 0.007$, $p < 0.001$) and high surprisal words elicited a larger reduction of N400 amplitude when meaningful gestures were present, in comparison to low surprisal words (experiment 1: $\beta = 0.007$, $p < 0.001$; experiment 2: $\beta = 0.011$, $p < 0.001$).

By contrast, we found a significant negative main effect of beat gestures (figure 4; experiment 1: $\beta = -0.004$, $p = 0.001$, accounting for 5% N400 change; experiment 2: $\beta = -0.006$, $p = 0.001$, accounting for 6% N400 change): words accompanied by beat gestures elicited a more negative N400. Moreover, high surprisal words accompanied by beat gestures showed even more negative N400 compared with low surprisal words (experiment 1: $\beta = -0.009$, $p < 0.001$, accounting for 12% N400 change; experiment 2: $\beta = -0.010$, $p < 0.001$, accounting for 10% N400 change).

## (c) What are the dynamics of multimodal cue processing?

We found significant interactions between multimodal cues (figure 5). First, we saw an interaction between pitch prosody (mean $F_0$) and meaningful gesture: words accompanied by meaningful gestures elicited even less negative N400 amplitude if their pitch prosody was higher (experiment 1: $\beta = 0.005$, $p < 0.001$, accounting for 4% N400 change; experiment 2: $\beta = 0.005$, $p < 0.001$). Second, the interactions between mouth informativeness and meaningful gesture (experiment 1: $\beta = 0.004$, $p = 0.002$; experiment 2: $\beta = 0.007$, $p < 0.001$) and between mouth informativeness and beat gesture (experiment 1: $\beta = 0.012$, $p < 0.001$; experiment 2: $\beta = 0.004$, $p = 0.001$) were also significant. Words with more informative mouth movement elicited less negative N400 when accompanied by either meaningful or beat gestures.

## 4. Discussion

We investigated for the first time the electrophysiological correlates of real-world multimodal language comprehension tracking the online processing as indexed by N400 amplitude. In real-world comprehension, listeners have access to gesture,

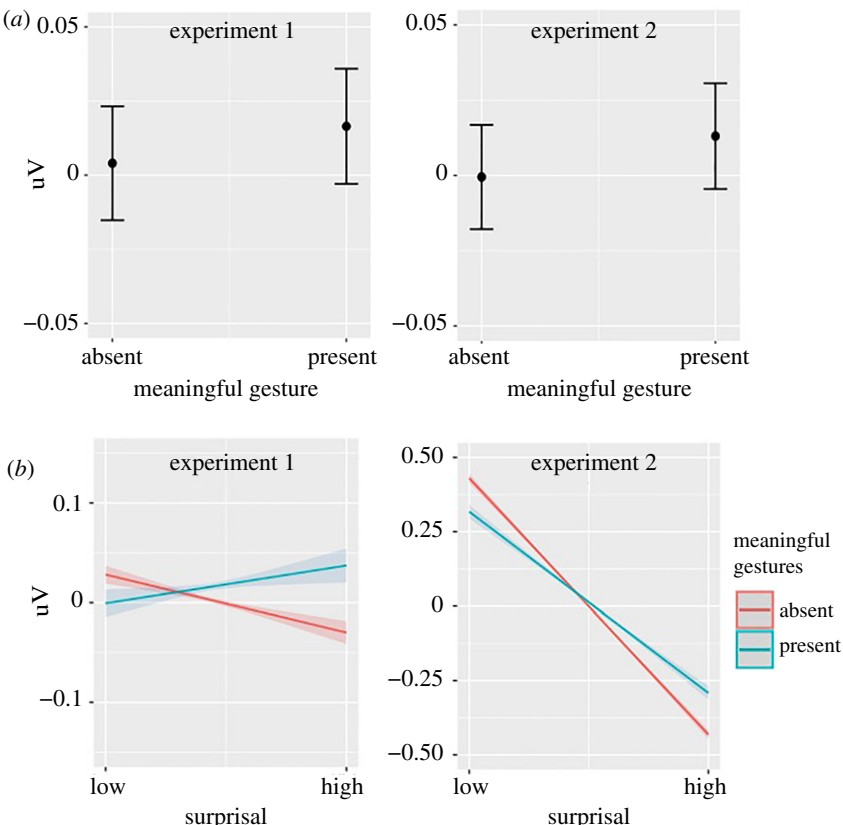

**Figure 3.** Meaningful gesture modulation of N400 amplitude. (a) Main effect of meaningful gestures. (b) Interaction between meaningful gestures and surprisal. (Online version in colour.)

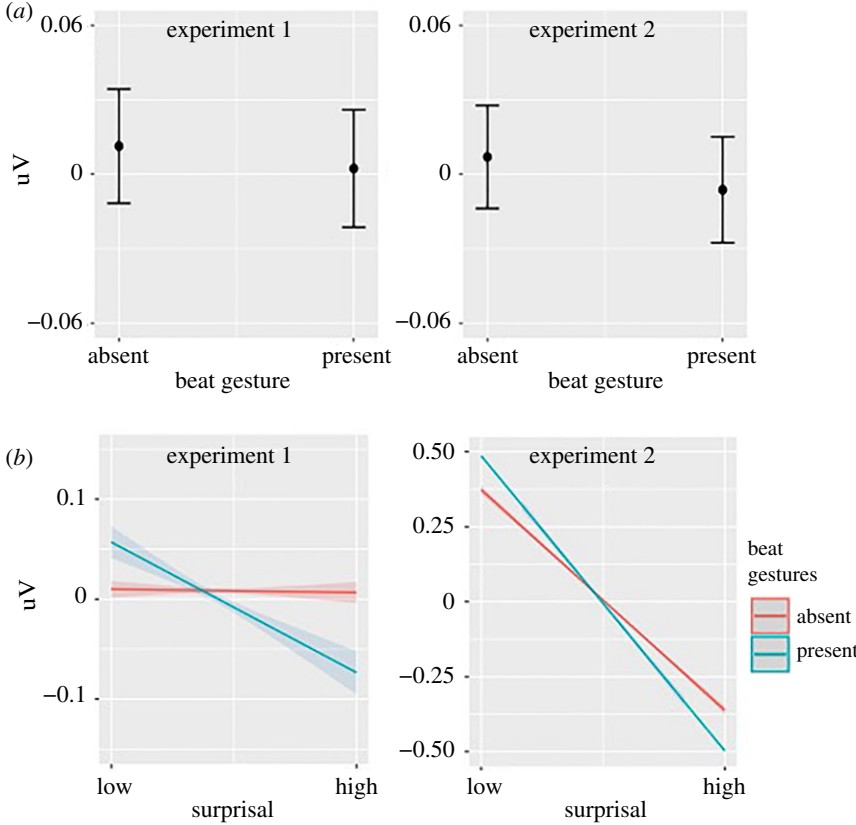

**Figure 4.** Beat gesture modulation of N400 amplitude. Main effect of beat gestures. (b) Interaction between beat gestures and surprisal. (Online version in colour.)

prosody and mouth movements. In contrast to previous studies that never used stimuli where all multimodal cues are present, we used (semi)naturalistic stimuli that did not break the naturally occurring correlations among multimodal cues. With these materials, we confirmed the N400 as a biomarker of prediction during naturalistic audiovisual

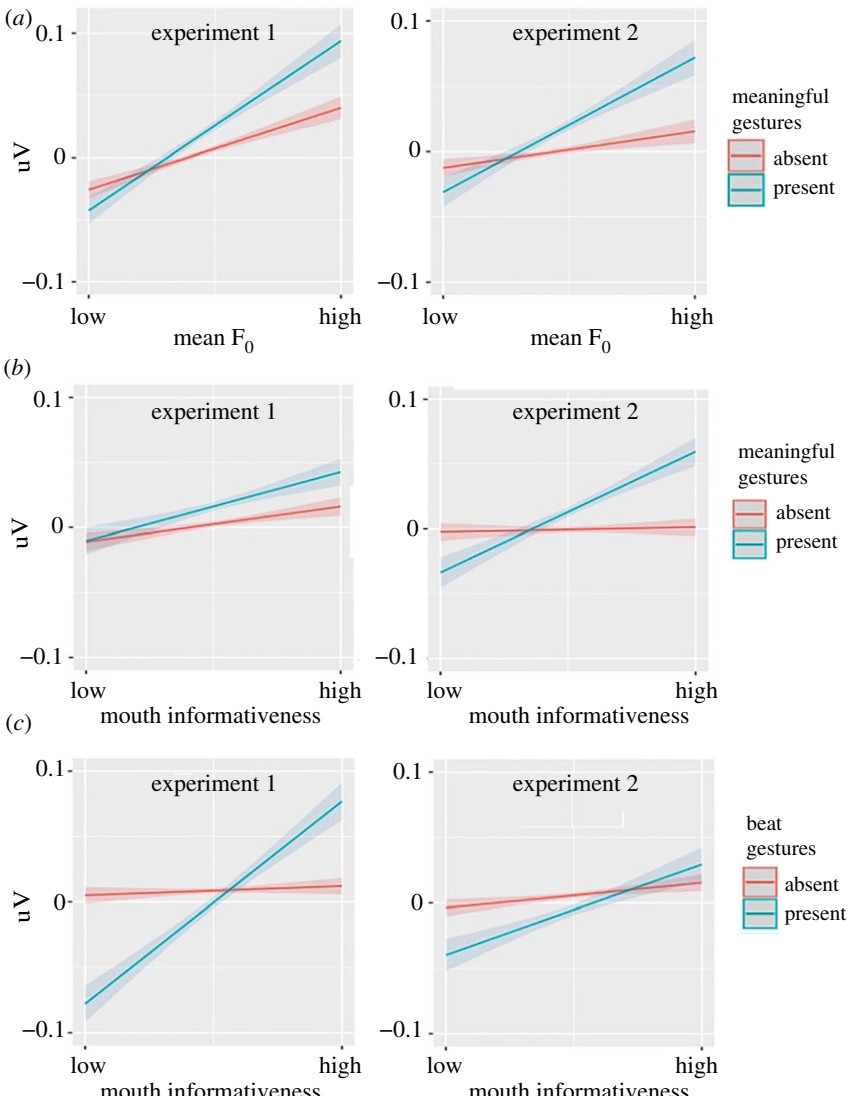

**Figure 5.** Interactions between multimodal cues. (*a*) Interaction between prosodic accentuation and meaningful gestures. (*b*) Interaction between meaningful gestures and mouth informativeness. (*c*) Interaction beat gestures and mouth informativeness. (Online version in colour.)

language comprehension: high surprisal words elicited a more negative N400 300–600 ms post-stimulus, strongest in the central-posterior electrodes.

We asked whether the processing of the multimodal cues is central to natural language processing. We address this question in an indirect manner, by assessing whether a key neurophysiological marker of prediction in language comprehension (N400) is modulated by the informativeness of the non-linguistic cues. We found that this is the case: each cue (except mouth informativeness) had a general effect and crucially modulated linguistic-based surprisal. Prosodic accentuation (pitch prosody) and meaningful gestures reduced the N400 amplitude overall, especially for high surprisal words. By contrast, the presence of beat gestures increased the N400 amplitude, especially for high surprisal words. Mouth movements did not modulate surprisal independently, but participated in complex interactions involving other cues. Thus, our results clearly show that prediction in language comprehension, in its natural face-to-face ecology, involves more than just linguistic information: the predictability of words based on linguistic context is *always* modulated by the multimodal cues thus forcing a reconsideration of theoretical claims strictly based on linguistic processing only [72].

Second, we addressed the dynamic nature of multimodal cue processing. We found that the weight given to each cue at any given time depends on which other cue is present, as indexed by interactions between cues. First, the facilitatory effect of meaningful gestures was enhanced with higher pitch prosody. Second, there is a facilitatory effect of mouth when gestures were present. Thus, investigating one cue at the time does not provide the full picture precluding the development of a mechanistic understanding of multimodal processing.

## (a) Prosody, gesture and mouth contributes to linguistic processing: beyond the state of the art

Using more naturalistic stimuli that do not artificially isolate single cues, our results provide needed clarification on previous conflicting results and important real-world generalization of previous results.

Prosodic accentuation is considered to mark 'newness' [6], as speakers are more likely to stress words conveying new information [24]. Electrophysiological studies have shown that unaccented new words—which represent an incongruence between prosodic pattern and information status—elicit more negative N400 [28,29]. Our findings complement previous work, showing that the presence of naturally

occurring accentuation (marked by higher pitch prosody) for less predictable words (but not more predictable words) leads to reduced N400 amplitude. We interpret this to indicate that linguistic context *and* prosody together determine the predictability of the next word: when linguistic context and prosody show the expected naturalistic correlation (high surprisal of word and high pitch) this is more predictable than when they do not. We found that meaningful gestures support processing, especially for high surprisal words. This is in line with studies that showed N400 reduction for the subordinate meaning of ambiguous words in the presence of a corresponding gesture [12], and previous work suggesting that incongruent gestures induce a larger N400 (see review [7]). Our results show that meaningful gestures play a more general role in face-to-face communication: they are always supporting word processing, not just in cases where processing is difficult due to incongruence or ambiguity.

Crucially, meaningful gestures, but not beat gestures, decrease the processing load in word processing, as indexed by smaller N400. This is probably because they can support prediction, given that production of meaningful gestures tend to start before production of their lexical affiliates [73]. High surprisal words accompanied by beat gestures elicited an even larger N400 effect. This effect may arise from beats enhancing the saliency of specific words [40], and highlighting its lack of fit into the previous context. Alternatively, listeners might try to extract meaning from all gestures and integrate it with speech by default. Since beats are not meaningful, integration fails, inducing processing difficulties. Importantly, the dissociation between meaningful and beat gestures further allows us to exclude the possibility that the N400 reduction observed (for meaningful gestures and prosody) resulted from cues sharing processing resources with speech, letting less predictable words go unnoticed.

Previous studies failed to find the same N400 effects of beat gestures [43,44]. However, these studies used artificial beat gestures (one single stroke per sentence), which are different from naturally occurring beat gestures (see also [74]). Alternatively, the lack of any meaningful gestures in previous studies could have discouraged listeners from paying attention to gestures. Shifts in the weight attributed to different cues based on specific tasks are documented in the literature [12,18,19] highlighting the importance of using ecologically valid paradigms.

Based on studies investigating single cues, it has been suggested that beats and prosodic accentuation serve the same function in communication, namely, making words more prominent [40]. However, our results provide evidence against such a claim as their electrophysiological correlates dissociated: beat gestures elicited more negative N400 especially for high surprisal words in line with the account above, while prosodic accentuation elicited less negative N400, especially for high surprisal words (see also [43]). Previous studies found a reliable N400 effect when prosodic accentuation mismatches with information status [28,29], while beat gestures interacts with information status in a later 600–900 ms time window, associated with meta-cognitive functions when processing general context [46].

We did not find a reliable effect of mouth informativeness as the main effect or in interaction with surprisal. Mouth movements have long been recognized to facilitate speech perception [47] and reduce early N1/P2 amplitude, indicating easier sensory-level processing [48]. However, our study focused on 300–600 ms to capture the effect of surprisal. Two previous studies have investigated the impact of mouth within the N400 time window. Hernández–Gutiérrez and colleagues did not find any N400 difference between audiovisual and audio-only speech [54]; while Brunellière and colleagues found an increase in N400 amplitude for more informative mouth movements [11]. Further research is necessary to clarify these discrepancies, however, our results suggest that mouth informativeness can affect processing in the N400 time window but only in combination with other cues in a multimodal context.

Finally, our results extend the previous literature by showing how cues interact. We found that meaningful gestures interact with prosody. Kristensen *et al.* argued that prosodic accentuation engages a domain general attention network [75]. Thus, accentuation may draw attention to other cues which consequently would be weighted more heavily. However, while plausible for meaningful gestures, it does not explain why similar enhancement for beat gestures is absent. Alternatively (or additionally) as argued by Holler and Levinson, listeners are attuned to natural correlations among the cues (e.g. high pitch correlates to larger mouth movements and increased gesture size) and would use cue-bundles for prediction [72]. Moreover, we found that the effect of mouth is enhanced whenever other visual cues (e.g. meaningful or beat gestures) are present. This may happen because mouth movements would fall within the focus of visual attention more easily if attention is already drawn to gestures (listeners look at the chin when processing sign language and gestures [76]).

## (b) Toward a neurobiology of natural communication

Our results call for neurobiological models of natural language comprehension, incorporating multimodal cues. In probabilistic-based predictive accounts, N400 is taken as an index of the processing demands associated with low predictability words [9]. It has been argued that prior to the bottom-up information, comprehenders hold a distribution of probabilistic hypotheses of the upcoming input constructed with prior knowledge and contextual information. This distribution, currently only based on linguistic input, is updated with new information, becoming the new prior distribution for the next event. Thus, N400 is linked to updating the distribution of hypotheses: smaller N400 is associated with more accurate prior distributions/predictions [9]. Our work shows that these mechanisms do not operate only on linguistic information, but crucially, they weigh in 'non-linguistic' multimodal cues.

Neuroanatomical models considering language in context and processed in interconnected networks [16,17] can, in principle, accommodate the results reported here. For example, in the Natural Organization of Language and Brain (NOLB) model, each multimodal cue is processed in different but partially overlapping sub-networks [17]. Indeed, different sub-networks have been associated with gestures and mouth movements, with a 'gesture network' and 'mouth network' weighted differently in different listening contexts [23,77]. These distributed sub-networks are assumed to provide constraints on possible interpretations of the acoustic signal, thus enabling fast and accurate comprehension [77]. Our finding of multiple interactions between cues is compatible with this view, thus suggesting that

multimodal prediction processes are dynamic, re-weighting each cue based on the status of other cues.

Ethics. All methods were carried out in accordance with relevant guidelines and regulations, and all experimental protocols were approved by the University College London ethics committee (0143/003).

Data accessibility. Data and scripts can be found here: https://osf.io/gzcya/?view_only=cb3ac98cd94d46bb9980c54a5fe72e4f.

Authors' contributions. Y.Z.: data curation, formal analysis, investigation, methodology, project administration, software, visualization, writing-original draft, writing-review and editing; D.F.: formal analysis, methodology, resources, software, validation; J.I.S.: validation, writing-original draft, writing-review and editing; J.T.: methodology, resources, supervision, validation; G.V.: conceptualization, funding acquisition, methodology, resources, supervision, validation, writing-original draft, writing-review and editing.

All authors gave final approval for publication and agreed to be held accountable for the work performed therein.

Competing interests. We declare we have no competing interests.

Funding. The work reported here was supported by a European Research Council Advanced Grant (ECOLANG, 743035) and Royal Society Wolfson Research Merit Award (WRM\R3\170016) to G.V.

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
