## [Peer Review File · Proceedings of the Royal Society B: Biological Sciences]

Review History

RSPB-2021-0500.R0 (Original submission)

Review form: Reviewer 1

Recommendation

Major revision is needed (please make suggestions in comments)

Scientific importance: Is the manuscript an original and important contribution to its field?

Excellent

General interest: Is the paper of sufficient general interest?

Good

Quality of the paper: Is the overall quality of the paper suitable?

Good

Is the length of the paper justified?

Yes

Should the paper be seen by a specialist statistical reviewer?

No

Do you have any concerns about statistical analyses in this paper? If so, please specify them explicitly in your report.

No

It is a condition of publication that authors make their supporting data, code and materials available - either as supplementary material or hosted in an external repository. Please rate, if applicable, the supporting data on the following criteria.

Is it accessible?

Yes

Is it clear?

Yes

Is it adequate?

Yes

Do you have any ethical concerns with this paper?

No

Comments to the Author

I very much like the article, and I think that this is the way for future research on language processing, traditionally studied as if it were not a between-humans communication system. The manuscript yields valuable results and has been performed proficiently, although there are some points that would need to be clarified. I will bring them in order:

-p. 5. Bottom. Citing other authors, it is here mentioned that accentuated information involves the left inferior frontal regions, related to phonological and semantic information. That region is also usually and typically related to syntactic processing, and indeed it is a pity that no mention of this domain appears, at least in the introduction, and whether some of the cues here studied could also affect linguistic processing in this domain. After all, the relationships between prosody and syntactic structure are well known.

-p. 6. Top. In the discussion on co-speech gestures, it would be nice if a classification of the existing types (i.e.: beats and meaningful) were established at the beginning.

-p. 6 Bottom. It seems convenient to comment on the functional implications of modulated brain responses in a window around 600 ms.

-p.6. Bottom. It would be nice if the functional linguistic processes presumably reflected by N1-P2 were mentioned.

-p.8. Material. It appears to me very important to state that the passages were unrelated (I guess this was the case). That is, they are not about the same topic or as in natural conversation. This is important, and relates to another contextual variable of potential value, though not studied here. Indeed, the presentation of the material would not be so natural in this regard, as human beings do not normally speak processing a number of unconnected sentences one after the other. Simply clarify.

-p. 8. Perhaps using only words likely to be accompanied by meaningful gestures would have been a better choice than doubly recording the sentences with/without gesturing, this making the comparisons comparable to those in the other variables.

-p.8. Bottom. Please clarify the specific meaning of 'informativeness' in this part of the text. Is it mouth informativeness? Probably not, but then there would be a terms confusion.

-p. 10. Mouth informativeness. Why weren't the same videos as used in the experimental sessions used for quantifying this? An edited zoom of the mouth would have worked, and more specific values for the actual experimental manipulations would be obtained.

-p. 10. Participants were instructed to watch videos carefully and answer the questions as quickly and accurately as possible: which questions? About what? Did they have to answer to every sentence? How many had a yes and how many a no as correct answer?

-p. 13 and others: the factor Prosody is sometimes called F0, and sometimes pitch. I find it more consistent with the naming of the other factors to call it prosody, even if in some occasions one

can use F0 to clarify or to avoid redundancies.

-p. 15. Discussion. Whether the explored cues are central to natural language processing is not studied here; only whether these cues may modulate this processing. Indeed, and this is a serious concern, as no performance data are reported (I ignore whether the yes/no questions used could help in this regard) we cannot assess whether language processing as such is importantly affected, apart from modulations of ERP waves related to predictability. This should be commented.

-p. 16. Top. As the effect of mouth was not significant as main effect, instead of assessing that the facilitatory effect of mouth was enhanced when gestures were present, it would be more correct to say that there is facilitatory effect of mouth when gestures were present.

-pp. 16-17. I find it a bit contradictory to say that meaningful gestures enhance expectation for lower probability continuations and, at the same time, that they decrease cognitive load in word processing. The link between one and the other assertion is not apparent, and indeed this is crucial to this paper. In this regard, are the modulations related specifically to word-language processes, or to general resources or general-domain processes. I can understand the link, and that this is not specifically the aim of the present paper, but this should be commented and clarified to better know the reach and consequences of the present data. Very important to this question is to clarify whether the gesture is simultaneous, preceding or following the word, and to which extent (in ms).

Review form: Reviewer 2 (Roel Willems)

Recommendation

Major revision is needed (please make suggestions in comments)

Scientific importance: Is the manuscript an original and important contribution to its field?

Excellent

General interest: Is the paper of sufficient general interest?

Good

Quality of the paper: Is the overall quality of the paper suitable?

Good

Is the length of the paper justified?

Yes

Should the paper be seen by a specialist statistical reviewer?

No

Do you have any concerns about statistical analyses in this paper? If so, please specify them explicitly in your report.

No

It is a condition of publication that authors make their supporting data, code and materials available - either as supplementary material or hosted in an external repository. Please rate, if applicable, the supporting data on the following criteria.

Is it accessible?

Yes

Is it clear?

Yes

Is it adequate?

Yes

Do you have any ethical concerns with this paper?

No

Comments to the Author

This article reports the results of two EEG experiments in which participants watched short movie clips of a person uttering language. The main goal of the paper is to investigate the separate and/or combined effects of multimodal cues on language comprehension. The study is part of growing literature calling for greater ecological validity in psycholinguistic studies. The paper is very novel and addresses an important topic in a creative manner. Most studies in the field focus on language comprehension without multimodal cues, and / or investigate single cues in isolation. This paper nicely breaks with those limitations. As such the study adds valuable information that will be of interest to a broad readership. The main comment is that crucial information is missing from the manuscript, and that the methods used need to be explained in more detail (see below).

I believe these issues can be clarified in a revised version of the manuscript. Below I give suggestions on how to improve the manuscript.

- Mainly, I am unsure of the exact task that the participants had to perform. I think they were meant to watch multiple videos with and without gestures and then had to answer yes/no questions to ensure they were paying attention. However, the authors mention including prosody in their research, but from how it is described, this aspect was assessed separately from the two EEG experiments. This concern also comes up with the analysis as the N400 was only analyzed without any combination of behavioral results. Please describe the task in more detail.
- Are the materials re-enactments of passages in the national corpus / BBC? This was unclear to me. Also, please provide more information on the characteristics of the stimuli, such as length per excerpt.
- Hierarchical linear modeling was used to determine the time window of interest in the EEG data. This is not a common procedure in the field and more information needs to be given on how this procedure works and why it is apt to use in this context.
- Is it fair to say that the paper is characterized as exploratory? The predictions are general. This is fine, but it should be made clearer whether the authors had specific a priori hypotheses or not.
- The introduction could provide more detail(s) and examples from previous research. At the moment, there are studies mentioned, but each is lacking more detail about how it links to the current study. This seems like something that could be easily be addressed.
- The addition of a figure that demonstrates the main tasks for each of the two experiments would greatly help with understanding the tasks and experiment as a whole
- The N for each experiment should be justified better. Additionally, there is a large difference in sample size between Exp. 1 and 2. Why is this?
- Regarding the coding of the gestures, details about the coding process are missing. What was done in cases of disagreement between the coders?
- Online rating experiment: the stimuli were presented outside of context here. Why was this done and does that make the ratings valuable for the main analysis?
- There is also a question about the multiple comparisons problem. If the study was of a more exploratory nature, then please specify this clearly.

Review form: Reviewer 3**Recommendation**

Accept with minor revision (please list in comments)

Scientific importance: Is the manuscript an original and important contribution to its field?
Good

General interest: Is the paper of sufficient general interest?
Good

Quality of the paper: Is the overall quality of the paper suitable?
Good

Is the length of the paper justified?
Yes

Should the paper be seen by a specialist statistical reviewer?
No

Do you have any concerns about statistical analyses in this paper? If so, please specify them explicitly in your report.
No

It is a condition of publication that authors make their supporting data, code and materials available - either as supplementary material or hosted in an external repository. Please rate, if applicable, the supporting data on the following criteria.

Is it accessible?
Yes

Is it clear?
Yes

Is it adequate?
Yes

Do you have any ethical concerns with this paper?
No

Comments to the Author
Summary

The authors report two ERP studies investigating the effects of non-linguistic cues (prosody, gestures, and mouth movements) on the magnitude of the N400 effect, which has been shown to correlate with surprisal in past work, during naturalistic audio-visual language comprehension. They find that prosody and meaningful gestures reduce the N400 (and more so for high-surprisal words), best gestures increase the N400, and mouth movements do not on their own affect the N400, but interact with other cues.

Evaluation

I enjoyed reading this paper. I am sympathetic to the need to evaluate language comprehension in more naturalistic, ecologically valid settings, where it is accompanied by the multitude of audio and visual cues, and I also appreciate the challenges that are inherent in such work. The studies that the authors report are well designed and the analyses are sound, and the results are compelling. I would support the publication of this manuscript, although the paper could be made stronger and clearer by addressing the comments below.

1. Although the materials are naturalistic, they are not fully natural. The authors should acknowledge this limitation.

2. The authors should be more careful in interpreting the results and acknowledge the possibility that their effects encompass lower-level (e.g., acoustic effects). For example, the authors write “words with higher mean F0 showed less negative EEG”. The topography of the N400 component is similar to the topography of auditory ERPs like the N1 P2, etc. Responses that originate in the auditory cortex will usually show as maximal around Cz. As a result, the finding of F0 affecting the EEG potential might be partly due to simple auditory responses to auditory frequency (further, pitch is usually correlated with loudness and louder sounds tend to cause stronger EEG responses).

They should also be clearer in their wording throughout. The title of one of the sections is “Multimodal cues are central to language processing: Modulation of word predictability by the cues”. Talking about modulating word predictability is imprecise and not justified. What the authors report are modulatory effects on the N400 ERP component. Although this component has been shown to correlate with word predictability, these are not equivalent / interchangeable. As noted above, N400 can also be affected by other factors that don’t have to do with word predictability given that the EEG scalp response is a sum of many processes happening at that time in the brain. Similarly, later, the authors write, “To assess the contribution of multimodal cues to the linguistic processing of a word measured as surprisal”. Again, this wording mixes between the dependent variable (the EEG potential) and a theoretical construct (and one of the independent variables, which is the linguistic surprisal of the words).

3. For statistical reporting. First, the authors should provide estimates of effect sizes, which would be easy for readers to interpret. For example, you could quantify the modulatory effects of prosody / meaningful gestures as % of the magnitude of the N400 (i.e., the size of the deviation from 0), so you could say e.g., that the effect of prosody on the N400 was 10% of the size of the N400 effect. Understanding how large these effects are is critical for their interpretation (with sufficient power, even tiny effects can be highly reliable). And second, for significance values, please specify the actual p-values (not just $p < 0.05$).

4. In a couple of places in the introduction and discussion, the authors draw on past fMRI evidence in a very reverse-inference (e.g., Poldrack, 2006) kind of way. E.g., they say “Such incongruence induces increased activation in the left inferior frontal gyrus, suggesting increased difficulty in phonological and semantic processing”. ‘Inferior frontal gyrus’ has been linked with every imaginable perceptual, motor, and cognitive function. Activity within this area cannot be used as a marker of a particular cognitive process like ‘phonological and semantic processing’. Given that this is not critical for you to mention, I would strongly recommend avoiding such statements. Further, the authors talk about evidence that gestures activate language-processing brain regions: “Meaningful gestures activate posterior middle-temporal and inferior frontal regions, associated with meaning processing across linguistic and non-linguistic materials”. The results of studies that have reported such overlap did not directly compare responses between language processing and gesture processing, and inferences about overlap rely on reverse-inference reasoning from coarse anatomical locations. Jouravlev et al. (2019, *Neuropsychologia*) use individual-subject analyses to show that the language-responsive areas do not actually respond to gestures. Again, these statements are not critical for anything in the current manuscript, so I would suggest omitting such references (or at least citing the relevant studies more comprehensively to acknowledge that this evidence has been challenged).

5. The authors write: “In the analyses we compare the same word across with/without gesture conditions because words likely to be accompanied by meaningful gestures (e.g., combing) are semantically very different from words that are not (e.g., pleasing).” Following the logic in 4, it would be interesting to separate the analysis of words that are more versus less likely to be accompanied by gestures. Are the reported effects larger for words that

are more naturally associated with gestures than words that are less?

6. The prosody cue is termed 'prosodic information', which is misleading. Information has a certain mathematical definition, associated with contextual predictability (as this paper discusses for linguistic predictability). But the prosody cue here is simply the F0 in Hz. Is it based on the assumption that the frequency/pitch in Hz is linearly associated with "prosodic surprise"? One could guess that a more relevant feature would be the pitch change relative to some expected baseline, and that in some instances a lower pitch can be more surprising than a higher pitch value. Perhaps raw pitch is good enough for a proxy of this. Nevertheless, I would encourage the authors refer to this cue as 'pitch prosody'. This would also make it clearer that the authors are not including other aspects of prosody like amplitude and timing.

Small additional comments / questions that would help clarify things:

Methods:

P. 8

- What is the nature of the corpus? Was it a spoken language corpus or written? (A spoken corpus would seem to fit best the purpose of this study.)

P. 8

The authors talk about recording two sets of videos: one where the actress was allowed to gesture as she normally would, and one where she was asked to hold her hands still. It might be informative to test how prosody and facial movements were affected by the lack of gesturing (i.e., did they become stronger to compensate? Or maybe the opposite?). The authors can assess this by comparing the amount of prosody and mouth movement between the different gesture conditions. (It is also unclear how these two conditions were handled in the analyses.)

P. 8-9

- Did word duration differ between words with/without gestures?

Figure 1:

The caption should provide more details about about the cues or refer to the methods.

- "Each frame corresponds to an image during each such word." This sentence is unclear.

P.10

- "reversed averaged phonological distance" Please provide a brief explanation of this measure in the text.

Exp 1

- "35 videos were followed by yes/no questions to ensure participants paid attention.."

It is unclear what these questions were. Were they comprehension questions?

- (See behavioural analysis in S.M.).

I would encourage to state all the important findings in the main text. E.g., was there a difference in behavioral responses to the different experimental conditions?

Exp 2

Here, the questions are referred to as "comprehension questions" Do you mean that these were like the yes/no questions but in an open answer form?

EEG analysis

Hierarchical linear modeling

- "Significant differences between the beta coefficient waveforms and zero ..." How was the significance determined?

- "0-1200ms time window was regressed against word surprisal." Each sample in this time

window? Or some averages across bins?

Linear Mixed Effect Regression Analysis

- Exp.1 performed linear..? Typo..? In Exp 1. we performed?
- "on the 300-600ms time window" I assume the voltage is averaged across this window?
- "We excluded from the analyses: (a) words without a surprisal value" Do you mean words that had a low surprisal value that didn't pass some threshold? Or the words that you didn't annotate to begin with?
- "(b) words without a mean F0 score" Please clarify: what does no mean F0 score mean?
- Explain clearly why do you exclude c) and d)

P.12

- Here the prosody variable name is switched to "mean F0". Inconsistent naming is unhelpful.
- "excluding any meaningful*beat gestures interactions". Why exclude those?

Just explain the logic behind those decisions

- "2) control: baseline, frequency, word length.." Frequency = unigram word frequency?
- "word order in the sentence" How was this calculated?
- "We further included word lemma and participant as random variables" How to include word lemma as a random variable? Do you mean just a list of all word lemmas?
- The maximal random structure failed to converge, so we included the highest interaction (threeway interactions) as random slope for participants⁷⁰, and surprisal as random slope for lemma."

This sentence is unclear. Did you mean "the highest interaction terms that converged"? It may be clearer to provide the LME model formula (e.g., in Wilkinson notation) for clarity.

- "...and 480,212 data points." What are these data points constructed of?
- "but not surprisal as random slope for item due to convergence issue." Does item = lemma here?

Results:

Figure 2

- A. Explain the color code of the figure - what are the blue/green areas?
- B. Is this for surprisal? Indicate what these figures show. Also, there are no units on the color bar.
- C. "red indicates the confidence interval." should be shaded pink because red is taken for the time window. "The red line underlying the figures" should be underlying the graphs.

Figure 3

- Consider showing some raw EEG traces also in these figures (from 3 on) to give a better sense of the data.
- From the graph, it seems like for high surprisal words, high F0 reduces the N400, which is discussed in the text, but for low surprisal words, high F0 enhances it, which is not discussed?
- How do you explain the difference between Exp. 1 and 2? It seems like in Exp 1 the surprisal effect on the N400 is reversed for high F0?

Discussion:

- "We found that ... each cue ... had a general effect and crucially modulated linguistic-based surprisal."
- Again (see p. 2 above), the cues did not modulate the surprisal, but the N400 or perhaps the effect of surprisal on the N400.
- "Thus, our results clearly show that language comprehension, in its natural face-to-face ecology, involves more than just speech."
- Because comprehension per se was not measured, but just an EEG measure that is associated

with it, it would be good to be more careful in the wording (e.g., 'language processing' is more neutral).

- "Prosody, gesture and mouth contributes to..." – a typo - contribute rather than contributes

- "Crucially, meaningful gestures, but not beat gestures, decrease the cognitive load in word processing." Unclear what is this based on. Talking about 'cognitive load' is much too general.

- "multimodal 'non-linguistic' cues have a central role in processing as they always modulate word predictability"

The authors use the word "always" throughout the paper; it is not clear what 'always' is used with respect to – either remove, or clearly specify the alternative.

Decision letter (RSPB-2021-0500.R0)

04-May-2021

Dear Mrs Zhang:

Your manuscript has now been peer reviewed and the reviews have been assessed by an Associate Editor. All of the reviewers, the AE, and I find your manuscript to be interesting and the topic of potential importance. However, the reviewers highlighted a number of issues that need to be addressed prior to further consideration, so I am inviting you to revise your manuscript. I will not repeat the reviewers' points here, but as you will see, most involve improving the details of the methods so that readers can better understand your procedure. In addition, I have one fairly minor point, which is that this issue (ecological validity and multimodal approaches in studying communication) is important for more than just human research; animal communication includes sounds as well as concurrent behaviors, gestures, or even facial expressions, yet often only one aspect is explored at a time. Given that the scope of the journal is to cover the breath of the biological sciences, it would be useful to mention this somewhere in your revised manuscript. The reviewers' comments (not including confidential comments to the Editor) and the comments from the Associate Editor are included at the end of this email for your reference.

To submit your revision please log into <http://mc.manuscriptcentral.com/prsb> and enter your Author Centre, where you will find your manuscript title listed under "Manuscripts with Decisions." Under "Actions", click on "Create a Revision" . Your manuscript number has been appended to denote a revision.

Research ethics:

Use of animals and field studies:

It is a condition of publication that you make available the data and research materials supporting the results in the article. Please see our Data Sharing Policies (<https://royalsociety.org/journals/authors/author-guidelines/#data>). Datasets should be deposited in an appropriate publicly available repository and details of the associated accession number, link or DOI to the datasets must be included in the Data Accessibility section of the article (<https://royalsociety.org/journals/ethics-policies/data-sharing-mining/>). Reference(s) to datasets should also be included in the reference list of the article with DOIs (where available).

Please submit a copy of your revised paper within three weeks. If we do not hear from you within this time your manuscript will be rejected. If you are unable to meet this deadline please let us know as soon as possible, as we may be able to grant a short extension.

Best wishes,
 Dr Sarah Brosnan
 Editor, Proceedings B
 mailto: proceedingsb@royalsociety.org

Associate Editor
 Board Member: 1
 Comments to Author:

The strength of this paper is in the novel and timely approach of using naturalistic stimuli to assess language comprehension. One reviewer notes that the stimuli is still not fully naturalistic, which should be noted clearly, but still the importance of this approach is rightly highlighted throughout the paper. All reviewers give helpful comments to make the methods clearer which will make the paper more accessible to a broad audience.

Reviewer(s)' Comments to Author:
 Referee: 1

Comments to the Author(s)

I very much like the article, and I think that this is the way for future research on language processing, traditionally studied as if it were not a between-humans communication system. The manuscript yields valuable results and has been performed proficiently, although there are some points that would need to be clarified. I will bring them in order:

-p. 5. Bottom. Citing other authors, it is here mentioned that accentuated information involves the left inferior frontal regions, related to phonological and semantic information. That region is also usually and typically related to syntactic processing, and indeed it is a pity that no mention of this domain appears, at least in the introduction, and whether some of the cues here studied could also affect linguistic processing in this domain. After all, the relationships between prosody and syntactic structure are well known.

-p. 6. Top. In the discussion on co-speech gestures, it would be nice if a classification of the existing types (i.e.: beats and meaningful) were established at the beginning.

-p. 6 Bottom. It seems convenient to comment on the functional implications of modulated brain responses in a window around 600 ms.

-p.6. Bottom. It would be nice if the functional linguistic processes presumably reflected by N1-P2 were mentioned.

-p.8. Material. It appears to me very important to state that the passages were unrelated (I guess this was the case). That is, they are not about the same topic or as in natural conversation. This is important, and relates to another contextual variable of potential value, though not studied here. Indeed, the presentation of the material would not be so natural in this regard, as human beings do not normally speak processing a number of unconnected sentences one after the other. Simply clarify.

-p. 8. Perhaps using only words likely to be accompanied by meaningful gestures would have been a better choice than doubly recording the sentences with/without gesturing, this making the comparisons comparable to those in the other variables.

-p.8. Bottom. Please clarify the specific meaning of 'informativeness' in this part of the text. Is it mouth informativeness? Probably not, but then there would be a terms confusion.

-p. 10. Mouth informativeness. Why weren't the same videos as used in the experimental sessions used for quantifying this? An edited zoom of the mouth would have worked, and more specific values for the actual experimental manipulations would be obtained.

-p. 10. Participants were instructed to watch videos carefully and answer the questions as quickly and accurately as possible: which questions? About what? Did they have to answer to every sentence? How many had a yes and how many a no as correct answer?

-p. 13 and others: the factor Prosody is sometimes called F0, and sometimes pitch. I find it more consistent with the naming of the other factors to call it prosody, even if in some occasions one can use F0 to clarify or to avoid redundancies.

-p. 15. Discussion. Whether the explored cues are central to natural language processing is not studied here; only whether these cues may modulate this processing. Indeed, and this is a serious concern, as no performance data are reported (I ignore whether the yes/no questions used could help in this regard) we cannot assess whether language processing as such is importantly affected, apart from modulations of ERP waves related to predictability. This should be commented.

-p. 16. Top. As the effect of mouth was not significant as main effect, instead of assessing that the facilitatory effect of mouth was enhanced when gestures were present, it would be more correct to say that there is facilitatory effect of mouth when gestures were present.

-pp. 16-17. I find it a bit contradictory to say that meaningful gestures enhance expectation for lower probability continuations and, at the same time, that they decrease cognitive load in word processing. The link between one and the other assertion is not apparent, and indeed this is crucial to this paper. In this regard, are the modulations related specifically to word-language processes, or to general resources or general-domain processes. I can understand the link, and that this is not specifically the aim of the present paper, but this should be commented and clarified to better know the reach and consequences of the present data. Very important to this question is to clarify whether the gesture is simultaneous, preceding or following the word, and to which extent (in ms).

Referee: 2

Comments to the Author(s)

This article reports the results of two EEG experiments in which participants watched short movie clips of a person uttering language. The main goal of the paper is to investigate the separate and/or combined effects of multimodal cues on language comprehension. The study is part of growing literature calling for greater ecological validity in psycholinguistic studies. The paper is very novel and addresses an important topic in a creative manner. Most studies in the field focus on language comprehension without multimodal cues, and / or investigate single cues in isolation. This paper nicely breaks with those limitations. As such the study adds valuable information that will be of interest to a broad readership. The main comment is that crucial information is missing from the manuscript, and that the methods used need to be explained in more detail (see below).

I believe these issues can be clarified in a revised version of the manuscript. Below I give suggestions on how to improve the manuscript.

- Mainly, I am unsure of the exact task that the participants had to perform. I think they were meant to watch multiple videos with and without gestures and then had to answer yes/no questions to ensure they were paying attention. However, the authors mention including prosody in their research, but from how it is described, this aspect was assessed separately from the two EEG experiments. This concern also comes up with the analysis as the N400 was only analyzed without any combination of behavioral results. Please describe the task in more detail.

- Are the materials re-enactments of passages in the national corpus / BBC? This was unclear to me. Also, please provide more information on the characteristics of the stimuli, such as length per excerpt.

-Hierarchical linear modeling was used to determine the time window of interest in the EEG data. This is not a common procedure in the field and more information needs to be given on how this procedure works and why it is apt to use in this context.

-Is it fair to say that the paper is characterized as exploratory? The predictions are general. This is fine, but it should be made clearer whether the authors had specific a priori hypotheses or not.

-The introduction could provide more detail(s) and examples from previous research. At the moment, there are studies mentioned, but each is lacking more detail about how it links to the current study. This seems like something that could be easily be addressed.

- The addition of a figure that demonstrates the main tasks for each of the two experiments would greatly help with understanding the tasks and experiment as a whole
- The N for each experiment should be justified better. Additionally, there is a large difference in sample size between Exp. 1 and 2. Why is this?
- Regarding the coding of the gestures, details about the coding process are missing. What was done in cases of disagreement between the coders?
- Online rating experiment: the stimuli were presented outside of context here. Why was this done and does that make the ratings valuable for the main analysis?
- There is also a question about the multiple comparisons problem. If the study was of a more exploratory nature, then please specify this clearly.

Referee: 3

Comments to the Author(s)

Summary

The authors report two ERP studies investigating the effects of non-linguistic cues (prosody, gestures, and mouth movements) on the magnitude of the N400 effect, which has been shown to correlate with surprisal in past work, during naturalistic audio-visual language comprehension. They find that prosody and meaningful gestures reduce the N400 (and more so for high-surprised words), best gestures increase the N400, and mouth movements do not on their own affect the N400, but interact with other cues.

Evaluation

I enjoyed reading this paper. I am sympathetic to the need to evaluate language comprehension in more naturalistic, ecologically valid settings, where it is accompanied by the multitude of audio and visual cues, and I also appreciate the challenges that are inherent in such work. The studies that the authors report are well designed and the analyses are sound, and the results are compelling. I would support the publication of this manuscript, although the paper could be made stronger and clearer by addressing the comments below.

1. Although the materials are naturalistic, they are not fully natural. The authors should acknowledge this limitation.
2. The authors should be more careful in interpreting the results and acknowledge the possibility that their effects encompass lower-level (e.g., acoustic effects). For example, the authors write "words with higher mean F0 showed less negative EEG". The topography of the N400 component is similar to the topography of auditory ERPs like the N1 P2, etc. Responses that originate in the auditory cortex will usually show as maximal around Cz. As a result, the finding of F0 affecting the EEG potential might be partly due to simple auditory responses to auditory frequency (further, pitch is usually correlated with loudness and louder sounds tend to cause stronger EEG responses).

They should also be clearer in their wording throughout. The title of one of the sections is "Multimodal cues are central to language processing: Modulation of word predictability by the cues". Talking about modulating word predictability is imprecise and not justified. What the authors report are modulatory effects on the N400 ERP component. Although this component has been shown to correlate with word predictability, these are not equivalent / interchangeable. As noted above, N400 can also be affected by other factors that don't have to do with word predictability given that the EEG scalp response is a sum of many processes happening at that time in the brain. Similarly, later, the authors write, "To assess the contribution of multimodal cues to the linguistic processing of a word measured as surprisal". Again, this wording mixes between the dependent variable (the EEG potential) and a theoretical construct (and one of the independent variables, which is the linguistic surprisal of the words).

3. For statistical reporting. First, the authors should provide estimates of effect sizes, which would be easy for readers to interpret. For example, you could quantify the modulatory effects of prosody / meaningful gestures as % of the magnitude of the N400 (i.e., the size of the deviation from 0), so you could say e.g., that the effect of prosody on the N400 was 10% of the size of the N400 effect. Understanding how large these effects are is critical for their interpretation (with sufficient power, even tiny effects can be highly reliable). And second, for significance values, please specify the actual p-values (not just $p < 0.05$).

4. In a couple of places in the introduction and discussion, the authors draw on past fMRI evidence in a very reverse-inference (e.g., Poldrack, 2006) kind of way. E.g., they say “Such incongruence induces increased activation in the left inferior frontal gyrus, suggesting increased difficulty in phonological and semantic processing”. ‘Inferior frontal gyrus’ has been linked with every imaginable perceptual, motor, and cognitive function. Activity within this area cannot be used as a marker of a particular cognitive process like ‘phonological and semantic processing’. Given that this is not critical for you to mention, I would strongly recommend avoiding such statements. Further, the authors talk about evidence that gestures activate language-processing brain regions: “Meaningful gestures activate posterior middle-temporal and inferior frontal regions, associated with meaning processing across linguistic and non-linguistic materials”. The results of studies that have reported such overlap did not directly compare responses between language processing and gesture processing, and inferences about overlap rely on reverse-inference reasoning from coarse anatomical locations. Jouravlev et al. (2019, *Neuropsychologia*) use individual-subject analyses to show that the language-responsive areas do not actually respond to gestures. Again, these statements are not critical for anything in the current manuscript, so I would suggest omitting such references (or at least citing the relevant studies more comprehensively to acknowledge that this evidence has been challenged).

5. The authors write: “In the analyses we compare the same word across with/without gesture conditions because words likely to be accompanied by meaningful gestures (e.g., *combing*) are semantically very different from words that are not (e.g., *pleasing*).” Following the logic in 4, it would be interesting to separate the analysis of words that are more versus less likely to be accompanied by gestures. Are the reported effects larger for words that are more naturally associated with gestures than words that are less?

6. The prosody cue is termed ‘prosodic information’, which is misleading. Information has a certain mathematical definition, associated with contextual predictability (as this paper discusses for linguistic predictability). But the prosody cue here is simply the F0 in Hz. Is it based on the assumption that the frequency/pitch in Hz is linearly associated with “prosodic surprise”? One could guess that a more relevant feature would be the pitch change relative to some expected baseline, and that in some instances a lower pitch can be more surprising than a higher pitch value. Perhaps raw pitch is good enough for a proxy of this. Nevertheless, I would encourage the authors refer to this cue as ‘pitch prosody’. This would also make it clearer that the authors are not including other aspects of prosody like amplitude and timing.

Small additional comments / questions that would help clarify things:

Methods:

P. 8

- What is the nature of the corpus? Was it a spoken language corpus or written? (A spoken corpus would seem to fit best the purpose of this study.)

P. 8

The authors talk about recording two sets of videos: one where the actress was allowed to gesture as she normally would, and one where she was asked to hold her hands still. It might be informative to test how prosody and facial movements were affected by the lack of gesturing (i.e.,

did they become stronger to compensate? Or maybe the opposite?). The authors can assess this by comparing the amount of prosody and mouth movement between the different gesture conditions. (It is also unclear how these two conditions were handled in the analyses.)

P. 8-9

- Did word duration differ between words with/without gestures?

Figure 1:

The caption should provide more details about about the cues or refer to the methods.

- "Each frame corresponds to an image during each such word." This sentence is unclear.

P.10

- "reversed averaged phonological distance" Please provide a brief explanation of this measure in the text.

Exp 1

- "35 videos were followed by yes/no questions to ensure participants paid attention.."

It is unclear what these questions were. Were they comprehension questions?

- (See behavioural analysis in S.M.).

I would encourage to state all the important findings in the main text. E.g., was there a difference in behavioral responses to the different experimental conditions?

Exp 2

Here, the questions are referred to as "comprehension questions" Do you mean that these were like the yes/no questions but in an open answer form?

EEG analysis

Hierarchical linear modeling

- "Significant differences between the beta coefficient waveforms and zero ..." How was the significance determined?

- "0-1200ms time window was regressed against word surprisal." Each sample in this time window? Or some averages across bins?

Linear Mixed Effect Regression Analysis

- Exp.1 performed linear.. Typo..? In Exp 1. we performed?

- "on the 300-600ms time window" I assume the voltage is averaged across this window?

- "We excluded from the analyses: (a) words without a surprisal value" Do you mean words that had a low surprisal value that didn't pass some threshold? Or the words that you didn't annotate to begin with?

- "(b) words without a mean F0 score" Please clarify: what does no mean F0 score mean?

- Explain clearly why do you exclude c) and d)

P.12

- Here the prosody variable name is switched to "mean F0". Inconsistent naming is unhelpful.

- "excluding any meaningful*beat gestures interactions". Why exclude those?

Just explain the logic behind those decisions

- "2) control: baseline, frequency, word length.." Frequency = unigram word frequency?

- "word order in the sentence" How was this calculated?

- "We further included word lemma and participant as random variables" How to include word lemma as a random variable? Do you mean just a list of all word lemmas?

- The maximal random structure failed to converge, so we included the highest interaction (threeway interactions) as random slope for participants70, and surprisal as random slope for lemma."

This sentence is unclear. Did you mean "the highest interaction terms that converged"? It may be clearer to provide the LME model formula (e.g., in Wilkinson notation) for clarity.

- "...and 480,212 data points." What are these data points constructed of?
- "but not surprisal as random slope for item due to convergence issue." Does item = lemma here?

Results:

Figure 2

- Explain the color code of the figure - what are the blue/green areas?
- Is this for surprisal? Indicate what these figures show. Also, there are no units on the color bar.
- "red indicates the confidence interval." should be shaded pink because red is taken for the time window. "The red line underlying the figures" should be underlying the graphs.

Figure 3

- Consider showing some raw EEG traces also in these figures (from 3 on) to give a better sense of the data.
- From the graph, it seems like for high surprisal words, high F0 reduces the N400, which is discussed in the text, but for low surprisal words, high F0 enhances it, which is not discussed?
- How do you explain the difference between Exp. 1 and 2? It seems like in Exp 1 the surprisal effect on the N400 is reversed for high F0?

Discussion:

- "We found that ... each cue ... had a general effect and crucially modulated linguistic-based surprisal."

Again (see p. 2 above), the cues did not modulate the surprisal, but the N400 or perhaps the effect of surprisal on the N400.

- "Thus, our results clearly show that language comprehension, in its natural face-to-face ecology, involves more than just speech:"

Because comprehension per se was not measured, but just an EEG measure that is associated with it, it would be good to be more careful in the wording (e.g., 'language processing' is more neutral).

- "Prosody, gesture and mouth contributes to..." – a typo - contribute rather than contributes

- "Crucially, meaningful gestures, but not beat gestures, decrease the cognitive load in word processing." Unclear what is this based on. Talking about 'cognitive load' is much too general.

- "multimodal 'non-linguistic' cues have a central role in processing as they always modulate word predictability"

The authors use the word "always" throughout the paper; it is not clear what 'always' is used with respect to – either remove, or clearly specify the alternative.

Author's Response to Decision Letter for (RSPB-2021-0500.R0)

See Appendix A.

RSPB-2021-0500.R1 (Revision)

Review form: Reviewer 1 (Manuel Martin-Loeches)

Recommendation

Accept as is

Scientific importance: Is the manuscript an original and important contribution to its field?

Good

General interest: Is the paper of sufficient general interest?

Good

Quality of the paper: Is the overall quality of the paper suitable?

Good

Is the length of the paper justified?

Yes

Should the paper be seen by a specialist statistical reviewer?

No

Do you have any concerns about statistical analyses in this paper? If so, please specify them explicitly in your report.

No

It is a condition of publication that authors make their supporting data, code and materials available - either as supplementary material or hosted in an external repository. Please rate, if applicable, the supporting data on the following criteria.

Is it accessible?

Yes

Is it clear?

Yes

Is it adequate?

Yes

Do you have any ethical concerns with this paper?

No

Comments to the Author

The authors have responded appropriately to my concerns.

Decision letter (RSPB-2021-0500.R1)

28-Jun-2021

Dear Mrs Zhang

I am pleased to inform you that your manuscript entitled "More than words: Word predictability, prosody, gesture and mouth movements in natural language comprehension" has been accepted for publication in Proceedings B.

Data Accessibility section

Open Access

Paper charges

Sincerely,

Dr Sarah Brosnan

Associate Editor:

Comments to Author:

The authors have done a fine job taking on the suggestions from the reviewers and the paper is improved. In my view, the finding will make a good contribution to the literature on multimodality in language comprehension.

Appendix A

Dear Prof Brosnan,

Thank you and the reviewers for the very helpful comments. Below, we spell out how we addressed them and how (and where) we modified the ms as a consequence.

You raised the concern that “this issue (ecological validity and multi-modal approaches in studying communication) is important for more than just human research; animal communication includes sounds as well as concurrent behaviors, gestures, or even facial expressions, yet often only one aspect is explored at a time. Given that the scope of the journal is to cover the breath of the biological sciences, it would be useful to mention this somewhere in your revised manuscript.”

We found this to be a really good point and discovered that the first two papers describing multimodal communication in monkeys were published in 2017. These are now cited, along with a couple of review papers looking at individual modalities (see p. 3).

Referee: 1

-p. 5. Bottom. Citing other authors, it is here mentioned that accentuated information involves the left inferior frontal regions, related to phonological and semantic information. That region is also usually and typically related to syntactic processing, and indeed it is a pity that no mention of this domain appears, at least in the introduction, and whether some of the cues here studied could also affect linguistic processing in this domain. After all, the relationships between prosody and syntactic structure are well known.

Because of space constraints, the fact that this point is not central to our arguments and following the suggestion by Reviewer 3, we now discuss the engagement of IIFG in terms of general processing difficulty rather than specifically semantics or prosody.

-p. 6. Top. In the discussion on co-speech gestures, it would be nice if a classification of the existing types (i.e.: beats and meaningful) were established at the beginning.

The classification is now added when we first introduce the different types of gestures on p. 5.

-p. 6 Bottom. It seems convenient to comment on the functional implications of modulated brain responses in a window around 600 ms.

Functional implications are now mentioned on p. 5.

-p.6. Bottom. It would be nice if the functional linguistic processes presumably reflected by N1-P2 were mentioned.

Functional implications are now mentioned on p. 5.

-p.8. Material. It appears to me very important to state that the passages were unrelated (I guess this was the case). That is, they are not about the same topic or as in natural conversation. This is important, and relates to another contextual variable of potential value, though not studied here. Indeed, the presentation of the material would not be so natural in this regard, as human beings do not normally speak processing a number of unconnected sentences one after the other. Simply clarify.

This is an important point. We now make clear in the description of our stimuli (Materials) that they are more naturalistic-like (rather than being naturalistic). While this is an important distinction, we do not believe that it changes greatly the type of co-occurrence and synchronization among the cues that we investigate here. One piece of evidence for this is that, as we discuss in the S.M., surprisal measures taken from different models varying with respect to the locality of their predictions did not differ in how well they could predict the EEG response.

-p. 8. Perhaps using only words likely to be accompanied by meaningful gestures would have been a better choice than doubly recording the sentences with/without gesturing, this making the comparisons comparable to those in the other variables.

Passages were randomly chosen from the BNC with the constraint that they include at least one gesturable verb in an attempt to have gesturable words as stimuli while at the same time ensuring the representativeness and naturalness of the materials. As the actress was not instructed where or how to gesture (again to preserve naturalness of the materials) it was really difficult to decide how to compare words with and without gestures. This is because, in contrast to prosodic modulation that is present on every word, for gestures there are many different reasons why some words are going to be gestured or not in a given context and, crucially it is nearly impossible to have naturalistic stimuli only containing gesturable words (even for completely concrete sentences -- e.g., walk to school everyday -- not all words are similarly gesturable. Thus, we decided that comparing the same words (presented in the same linguistic context, thus with the same surprisal) would provide the most rigorous way to assess the impact of gestures on word surprisal.

-p.8. Bottom. Please clarify the specific meaning of 'informativeness' in this part of the text. Is it mouth informativeness? Probably not, but then there would be a terms confusion.

This has been changed.

-p. 10. Mouth informativeness. Why weren't the same videos as used in the experimental sessions used for quantifying this? An edited zoom of the mouth would have worked, and more specific values for the actual experimental manipulations would be obtained.

Our goal was to use a robust measure quantified over word-types rather than word-tokens embedded in connected speech. While we agree with the reviewer that using measures derived from the passages could have provided a better fit, our method ensures greater generalizability of the results. Moreover, the norms we developed for the study can be used in other studies and shared across researchers.

-p. 10. Participants were instructed to watch videos carefully and answer the questions as quickly and accurately as possible: which questions? About what? Did they have to answer to every sentence? How many had a yes and how many a no as correct answer?

The questions were comprehension questions about the content of immediately preceding passages (e.g. Passage: "Emma screamed and swore at them. She was especially angry if the girls dared to eat any of her food or drink her coffee.", Question: "Is Emma going to share her sweets with the other girls?"). In Exp.1, we randomly chose 35 passages to be followed by questions (35% of all the passages, randomly distributed in the experiment, 14 questions had a yes answer and 21 a no answer). In Exp. 2, we included 40 questions (50% of the passages,

randomly distributed, 20 questions had a yes answer and 20 a no answer). For experiment 1, the questions were included in order to obtain a behavioural response. This was important because given the novelty of this experiment, we were unsure whether any behavioural effect of surprisal could be found. For both experiments, the questions were included to ensure that the participants are paying attention and understanding the materials. This information is now included in the S.M.

-p. 13 and others: the factor Prosody is sometimes called F0, and sometimes pitch. I find it more consistent with the naming of the other factors to call it prosody, even if in some occasions one can use F0 to clarify or to avoid redundancies.

Now called pitch prosody throughout (as also suggested by reviewer 3)

-p. 15. Discussion. Whether the explored cues are central to natural language processing is not studied here; only whether these cues may modulate this processing. Indeed, and this is a serious concern, as no performance data are reported (I ignore whether the yes/no questions used could help in this regard) we cannot assess whether language processing as such is importantly affected, apart from modulations of ERP waves related to predictability. This should be commented.

There is now substantial evidence that language processing is a predictive process (see review in Kuperberg & Jaeger, 2016). We find that the predictability of a word given its preceding linguistic context -- its lexical surprisal -- modulates N400. Although the comprehension questions were not introduced in order to provide a behavioural measure, we found that surprisal also modulated RTs in Experiment 1. This effect was non-significant in Experiment 2 most likely because passages as well as the comprehension questions were much longer.

The multimodal non-linguistic cues we investigate here are usually not considered in studies of language processing. We could have found that they do not modulate linguistic-based prediction, or that they modulate it only in some cases or only some of the cues. However, this is not what we found: we see in the study that all of them are used and they critically interact with the linguistic-based surprisal thus changing the neural signatures of linguistic predictability. This to us is close enough to indicate that the cues are central to the processing. We have clarified this in the discussion, see p. 15.

-p. 16. Top. As the effect of mouth was not significant as main effect, instead of assessing that the facilitatory effect of mouth was enhanced when gestures were present, it would be more correct to say that there is facilitatory effect of mouth when gestures were present.

We have now updated this, see pg 15.

-pp. 16-17. I find it a bit contradictory to say that meaningful gestures enhance expectation for lower probability continuations and, at the same time, that they decrease cognitive load in word processing. They link between one and the other assertion is not apparent, and indeed this is crucial to this paper. In this regard, are the modulations related specifically to word-language processes, or to general resources or general-domain processes. I can understand the link, and that this is not specifically the aim of the present paper, but this should be commented and clarified to better know the reach and consequences of the present data. Very important to this question is to clarify whether the gesture is simultaneous, preceding or following the word, and to which extent (in ms).

We speculate in the ms that meaningful gestures affect processing by virtue of making the upcoming word (the lexical affiliate of the gesture) more predictable. Although we did not measure this in the current study, the time alignment between the beginning of the meaningful gesture and the beginning of the word, recent corpus-based evidence by Ter Bekke (2021) indicates that gestures start before the word (in 96% of the instances in the corpus). And, moreover, often (62% of the instances) the most informative part of the gesture (its stroke) is completed before the word. As they increase the predictability of words that otherwise would be unpredictable, then meaningful gestures would decrease processing load. The situation is very different for beat gestures which instead would highlight specific unpredictable words making them stand out even more and therefore increasing the processing load. We have clarified our hypothesised mechanism for meaningful gestures on p. 16.

Referee: 2

- Mainly, I am unsure of the exact task that the participants had to perform. I think they were meant to watch multiple videos with and without gestures and then had to answer yes/no questions to ensure they were paying attention. However, the authors mention including prosody in their research, but from how it is described, this aspect was assessed separately from the two EEG experiments. This concern also comes up with the analysis as the N400 was only analyzed without any combination of behavioral results. Please describe the task in more detail.

We have now rewritten the description of the task on p. 8-10 and in the S.M.. In both experiments, participants watched videoclips of a female speaker producing short passages while their EEG was recorded. Immediately after some of the passages ($\frac{1}{3}$ for Exp.1 and $\frac{1}{2}$ for Exp.2), participants were then asked to answer comprehension questions about the content of the passages (see additional details in our response to Reviewer 1). We prepared two versions for each passage: in one the actress was free to produce gestures, in the other we asked her not to produce any gesture. Half of the passages presented to participants had gestures, and half had no gestures (the gesture and no-gesture versions of the same passage were presented to different participants, see above our response to Reviewer 1 for an explanation of why we manipulated gestures in this way).

Prosody was measured for each (content) word in each passage in Exp 1 and Exp 2.

We did analyse behavioural results from the comprehension questions in Experiment 1 (experiment 2 used passages that were too long to make analyses of RTs for the questions meaningful) to ensure that our measures of surprisal had a behavioural effect in addition to the electrophysiological effect indexed by the N400.

- Are the materials re-enactments of passages in the national corpus / BBC? This was unclear to me. Also, please provide more information on the characteristics of the stimuli, such as length per excerpt.

Yes, the materials are re-enactment of BNC & BBC corpuses.

In Exp.1, the averaged number of word is 23, the averaged duration of the videos is 8.50s. In Exp.2, the averaged number of word is 45, the averaged duration of the videos is 15.66s. The above information is now reported in the S.M.

-Hierarchical linear modeling was used to determine the time window of interest in the EEG data. This is not a common procedure in the field and more information needs to be given on how this procedure works and why it is apt to use in this context.

Hierarchical linear modelling is used to determine the time window sensitive to surprisal, as no previous study investigated surprisal in audiovisual multimodal communication. We selected hierarchical linear modelling, instead of more traditional mass Univariate approaches or simple visual inspection, because hierarchical linear modelling can better accommodate continuous variables (surprisal here). Hierarchical linear modelling (LIMO toolbox) is one way of carrying out regression based EEG analysis (Smith & Kutas, 2015 a, b). It decomposes the ERP signal into a time-series of beta coefficient waveforms elicited by continuous variables. Significant differences between the beta coefficient waveforms and zero (or a flat line, indicating that the variable does not affect EEG signal) represent the existence of an effect. Similar regression based approaches have been successfully used in previous EEG studies investigating other continuous variables (e.g. Rousselet et al., 2011; Broderick et al., 2018).

In this analysis, we first created a single-trial file from the EEG file for each participant, and a continuous variable containing surprisal of each word that the specific participant was presented with. In the first level analysis for each participant, a regression analysis was performed for each data point (sample, based on sampling rate, which is 512Hz in our case) in 0-1200ms time window per electrode per word, with EEG voltage as the dependent variable and word surprisal as the independent variable, thus generating a matrix of beta values, which indicate whether and when surprisal has an effect for each participant. In the second level of the analysis across all participants, the averaged beta matrix was compared with 0 using a one- sample t-test (bootstrap set at 1000, clustering corrected against spatial and temporal multiple comparison, Pernet et al., 2015). The resulting significant time window represents the interval where surprisal reliability modulates the EEG response.

This has been added to the S.M..

-Is it fair to say that the paper is characterized as exploratory? The predictions are general. This is fine, but it should be made clearer whether the authors had specific a priori hypotheses or not.

As we now spell out on p. 7, we have a priori predictions with respect to our first question (whether the cues are central to language processing); however, not with respect to our second question (whether/how they interact).

-The introduction could provide more detail(s) and examples from previous research. At the moment, there are studies mentioned, but each is lacking more detail about how it links to the current study. This seems like something that could be easily be addressed.

We try now to make the links more transparent.

-The addition of a figure that demonstrates the main tasks for each of the two experiments would greatly help with understanding the tasks and experiment as a whole

Figure 1 is now updated to illustrate both the tasks and the quantification of cues. See p.7.

-The N for each experiment should be justified better. Additionally, there is a large difference in sample size between Exp. 1 and 2. Why is this?

Exp 1: N was decided on the basis of the previous study by Frank et al (2015). They used 24 subjects and found a significant effect of surprisal (computed as in the present paper) in word-by-word reading. We decided to increase N because of the difference in presentation modality.

Exp2 has longer passages (on average 45 words per passage compared with 23 in Exp.1). Therefore, we are able to obtain a similar number of observations with less participants. This information is now added in S.M.

- Regarding the coding of the gestures, details about the coding process are missing. What was done in cases of disagreement between the coders?

Details of the coding are provided in the S.M.. In case of disagreements, coders discuss the rational and/or consult other expert coders. Once an agreement was reached, the annotation was updated.

-Online rating experiment: the stimuli were presented outside of context here. Why was this done and does that make the ratings valuable for the main analysis?

See our response to Reviewer 1 above.

-There is also a question about the multiple comparisons problem. If the study was of a more exploratory nature, then please specify this clearly.

Only one model was run per each study therefore there were no multiple comparisons.

Referee: 3

1. Although the materials are naturalistic, they are not fully natural. The authors should acknowledge this limitation.

This is now stated in the Materials section.

2. The authors should be more careful in interpreting the results and acknowledge the possibility that their effects encompass lower-level (e.g., acoustic effects). For example, the authors write “words with higher mean F0 showed less negative EEG”. The topography of the N400 component is similar to the topography of auditory ERPs like the N1 P2, etc. Responses that originate in the auditory cortex will usually show as maximal around Cz. As a result, the finding of F0 affecting the EEG potential might be partly due to simple auditory responses to auditory frequency (further, pitch is usually correlated with loudness and louder sounds tend to cause stronger EEG responses).

We believe the effect of pitch/F0 is unlikely to be purely acoustic, because it was found within 300-600ms, which is too late for acoustic processes, and because the F0 effect interacts with processing of other information (e.g. linguistic surprisal).

They should also be clearer in their wording throughout. The title of one of the sections is “Multimodal cues are central to language processing: Modulation of word predictability by the cues”. Talking about

modulating word predictability is imprecise and not justified. What the authors report are modulatory effects on the N400 ERP component. Although this component has been shown to correlate with word predictability, these are not equivalent / interchangeable. As noted above, N400 can also be affected by other factors that don't have to do with word predictability given that the EEG scalp response is a sum of many processes happening at that time in the brain. Similarly, later, the authors write, "To assess the contribution of multimodal cues to the linguistic processing of a word measured as surprisal". Again, this wording mixes between the dependent variable (the EEG potential) and a theoretical construct (and one of the independent variables, which is the linguistic surprisal of the words).

We are using more careful wording throughout. It is also worth noting that we are assessing whether the multimodal cues affect the N400 to words by virtue of interacting with the word's surprisal.

3. For statistical reporting. First, the authors should provide estimates of effect sizes, which would be easy for readers to interpret. For example, you could quantify the modulatory effects of prosody / meaningful gestures as % of the magnitude of the N400 (i.e., the size of the deviation from 0), so you could say e.g., that the effect of prosody on the N400 was 10% of the size of the N400 effect. Understanding how large these effects are is critical for their interpretation (with sufficient power, even tiny effects can be highly reliable). And second, for significance values, please specify the actual p-values (not just $p < 0.05$).

The effect size information is now derived from the standardized regression coefficient and added in the result section. The actual p-values are now specified in the ms.

4. In a couple of places in the introduction and discussion, the authors draw on past fMRI evidence in a very reverse-inference (e.g., Poldrack, 2006) kind of way. E.g., they say "Such incongruence induces increased activation in the left inferior frontal gyrus, suggesting increased difficulty in phonological and semantic processing". 'Inferior frontal gyrus' has been linked with every imaginable perceptual, motor, and cognitive function. Activity within this area cannot be used as a marker of a particular cognitive process like 'phonological and semantic processing'. Given that this is not critical for you to mention, I would strongly recommend avoiding such statements. Further, the authors talk about evidence that gestures activate language-processing brain regions: "Meaningful gestures activate posterior middle-temporal and inferior frontal regions, associated with meaning processing across linguistic and non-linguistic materials". The results of studies that have reported such overlap did not directly compare responses between language processing and gesture processing, and inferences about overlap rely on reverse-inference reasoning from coarse anatomical locations. Jouravlev et al. (2019, *Neuropsychologia*) use individual-subject analyses to show that the language-responsive areas do not actually respond to gestures. Again, these statements are not critical for anything in the current manuscript, so I would suggest omitting such references (or at least citing the relevant studies more comprehensively to acknowledge that this evidence has been challenged).

These statements have been amended. With respect to the correlation between incongruent prosody and IFG, we discuss this now in terms of processing difficulty without explicit reference to phonology or semantics (p. 5). With respect to the discussion on gestures, we now present a more balanced discussion citing Jouravlev as suggested (p. 5).

5. The authors write: "In the analyses we compare the same word across with/without gesture conditions because words likely to be accompanied by meaningful gestures (e.g., combing) are semantically very different from words that are not (e.g., pleasing)."

Following the logic in 4, it would be interesting to separate the analysis of words that are more versus less likely to be accompanied by gestures. Are the reported effects larger for words that are more naturally associated with gestures than words that are less?

This seems like an interesting idea, however, we decided not to pursue it in the present ms for the following reasons. First, words may be more or less gesturable (where gesturable here refers to meaning gestures) for different reasons and this may have different processing implications. So, less concrete words are less gesturable, but also concrete words like sea, sun are not really gesturable despite having clear sensory features. Second, and perhaps more important, we find it difficult to think why there would be an interaction between gesturability and surprisal.

6. The prosody cue is termed 'prosodic information', which is misleading. Information has a certain mathematical definition, associated with contextual predictability (as this paper discusses for linguistic predictability). But the prosody cue here is simply the F0 in Hz. Is it based on the assumption that the frequency/pitch in Hz is linearly associated with "prosodic surprise"? One could guess that a more relevant feature would be the pitch change relative to some expected baseline, and that in some instances a lower pitch can be more surprising than a higher pitch value. Perhaps raw pitch is good enough for a proxy of this. Nevertheless, I would encourage the authors refer to this cue as 'pitch prosody'. This would also make it clearer that the authors are not including other aspects of prosody like amplitude and timing.

We are now using "pitch prosody" throughout (as also suggested by reviewer 1).

Small additional comments / questions that would help clarify things:

Methods:

P. 8

- What is the nature of the corpus? Was it a spoken language corpus or written? (A spoken corpus would seem to fit best the purpose of this study.)

In Experiment 1, we selected materials from the BNC (British National Corpus). In particular, we used the BNC (and not a web corpus) because it offers more standard sentences and the sentence order is preserved. BNC contains 100 million words of language material selected from both written (90%) and spoken language (10%). Our stimuli were taken from the written part of the corpus. Thus, in Experiment 2, in order to further enhance the naturalness of the stimuli, we selected passages from the BBC script library, containing scripts of BBC TV shows. Therefore, materials used in Exp.2 represent spoken language. This information is now reported in the S.M..

P. 8

The authors talk about recording two sets of videos: one where the actress was allowed to gesture as she normally would, and one where she was asked to hold her hands still. It might be informative to test how prosody and facial movements were affected by the lack of gesturing (i.e., did they become stronger to compensate? Or maybe the opposite?). The authors can assess this by comparing the amount of prosody and mouth movement between the different gesture conditions. (It is also unclear how these two conditions were handled in the analyses.)

Pitch prosody tends to be higher in videos with gesture than without (Exp.1 averaged F0: no gesture videos = 295.90Hz, gesture videos = 300.79Hz; Exp.2 averaged F0: no gesture videos =

268.71Hz, gesture videos = 282.84Hz). We cannot assess whether the mouth informativeness differs across gesture conditions, as mouth informativeness is measured separately in a single word recognition task, which is independent from the current study. These are now reported in S.M.

The presence of each type of gesture (meaningful & beat) are implemented in the analysis as two separate categorical variables. For both variables, the gesture present items come from the with gesture videos, and the gesture absent items come from the without gesture videos.

P. 8-9

- Did word duration differ between words with/without gestures?

Words with gestures tend to be ~15ms longer than words without (Exp.1 duration: no gesture videos=437.28ms, gesture videos=455.12ms; Exp.2 duration: no gesture videos=328.47ms, gesture videos=343.94ms). These differences are very small and we do not believe may impact our results. These are now reported in the S.M.

Figure 1:

The caption should provide more details about about the cues or refer to the methods.

- "Each frame corresponds to an image during each such word." This sentence is unclear.

Figure 1 has been changed in response to another Reviewer and the caption updated.

P.10

- "reversed averaged phonological distance" Please provide a brief explanation of this measure in the text.

Phonological distance measures the difference between participants' response with the target answer. Longer distance means less accurate guess, while 0 means a perfect guess. In order to measure how informative the mouth movement is, we then reversed the distance score by multiplying it with -1, so that smaller value means less informative mouth movement. This is now clarified in the S.M..

Exp 1

- "35 videos were followed by yes/no questions to ensure participants paid attention.."

It is unclear what these questions were. Were they comprehension questions?

Yes they were comprehension questions. For example, for the passage "Emma screamed and swore at them. She was especially angry if the girls dared to eat any of her food or drink her coffee", the question was "Is Emma going to share her sweets with the other girls?". This is clarified in p.10 and the S.M..

I would encourage to state all the important findings in the main text. E.g., was there a difference in behavioral responses to the different experimental conditions?

We decided not to report the behavioural results in the main text because of space constraints and because the comprehension task was originally introduced as an attention check, rather than as a way to obtain behavioural data on surprisal.

Exp 2

Here, the questions are referred to as “comprehension questions” Do you mean that these were like the yes/no questions but in an open answer form?

These were just as Experiment 1. This has now been rephrased in the ms (p. 10).

EEG analysis

Hierarchical linear modeling

- “Significant differences between the beta coefficient waveforms and zero ...” How was the significance determined?

Beta coefficient (averaged across participants) is compared with 0 using a one- sample t-test (bootstrap set at 1000, clustering corrected against spatial and temporal multiple comparison). Further information about hierarchical linear modeling is now added to S.M. as requested by reviewer 2.

- “0-1200ms time window was regressed against word surprisal.” Each sample in this time window? Or some averages across bins?

For each sample in this time window (based on the current sampling rate, which is 512 Hz). Now added to the S.M..

Linear Mixed Effect Regression Analysis

- Exp.1 performed linear..” Typo..? In Exp 1. we performed?

This has been Fixed.

- “on the 300-600ms time window” I assume the voltage is averaged across this window?

Yes. This is clarified in p.11.

- “We excluded from the analyses: (a) words without a surprisal value” Do you mean words that had a low surprisal value that didn't pass some threshold? Or the words that you didn't annotate to begin with?

Very few words do not have any surprisal value (Exp.1: n=9; Exp.2: n=13), due to the lack of co-occurrence between this word and its context in the training corpus. These further clarifications (the current one and below) have now been added to S.M..

- “(b) words without a mean F0 score” Please clarify: what does no mean F0 score mean?

Very few words (Exp.1: n=4; Exp.2: n=2) showed pitch error when using praat to automatically extract pitch (e.g. when the vowel is pronounced very quietly). Now clarified in the S.M..

- Explain clearly why do you exclude c) and d)

We excluded c) words with both meaningful and beat gesture (Exp.1: n=3; Exp.2: n=6). These instances usually represent the speaker producing a meaningful gesture but then a quick beat gesture for a word, or using one hand to produce a meaningful gesture but the other to produce a

beat. Given the rarity of this phenomenon, we excluded them from the analysis thus removing any interaction between meaningful and beat gestures.

We excluded in d) words occurring without any gesture in the “with gesture” condition, and the corresponding words in without gesture videos (Exp.1: n=406; Exp.2: n=685). This is to reduce the imbalance of the data, otherwise the without gesture condition would include not only the corresponding without gesture words of all with gesture words, but also all words in d), making this group ~3 times larger than the with gesture one.

Now clarified in the S.M..

P.12

- Here the prosody variable name is switched to “mean F0”. Inconsistent naming is unhelpful.

We now use pitch prosody throughout (as also suggested by reviewer 1).

- “excluding any meaningful*beat gestures interactions”. Why exclude those?

As explained above and now incorporated in the S.M., the co-occurrence between meaningful and beat gestures is so rare as to make this interaction uninformative.

Just explain the logic behind those decisions

- “2) control: baseline, frequency, word length..” Frequency = unigram word frequency?

We initially included ENCOW frequency in the model as control variable, but then removed it from the final model due to multiple collinearity with surprisal, as both measures capture linguistic probability to different extents. Frequency is now removed from manuscript p.11, and justification has been added to S.M..

- “word order in the sentence” How was this calculated?

“Word order in the sentence” is defined as the position of the current word in the sentence. E.g. in “Emma screamed ...”, word order of “scream” is 2. This is clarified in the S.M..

- “We further included word lemma and participant as random variables” How to include word lemma as a random variable? Do you mean just a list of all word lemmas?

For each word, its lemma is extracted (e.g. lemma “work” includes both word “worked” and “working”) and subsequently included as random intercept. Therefore, the model assumes each lemma may have a different intercept. The inclusion of by participant and item intercept (and slope when possible) is the traditional setup for such models when dealing with psycholinguistic studies (Barr et al., 2013). This is now clarified in the S.M..

- The maximal random structure failed to converge, so we included the highest interaction (threeway interactions) as random slope for participants⁷⁰, and surprisal as random slope for lemma.”
This sentence is unclear. Did you mean “the highest interaction terms that converged”? It may be clearer to provide the LME model formula (e.g., in Wilkinson notation) for clarity.

We added three way interactions (which is the highest interaction terms in our model) as random slope, as suggested by Barr, 2013. Full model formula is added in S.M..

- "...and 480,212 data points." What are these data points constructed of?

These data points are constructed of participants * words * electrode positions. For example, in Exp.1, there are 31 participants, 563 words (but grouped into 381 lemmas), and 32 electrodes, creating a maximum of 558,496 data points. However, some words may not be included in the analysis (e.g. ERP from some electrodes may be contaminated with noise and thus rejected in preprocessing), thus resulting in 480,212 data points. This is now added to S.M..

- "but not surprisal as random slope for item due to convergence issue." Does item = lemma here?

Yes. We changed the manuscript to avoid confusion.

Results:

Figure 2

A. Explain the color code of the figure - what are the blue/green areas?

The blue/green areas represent a significant negative beta value (compared with 0) across all participants and words. The timing differs across electrodes. Clarification is now added to the caption. Please see the changes in S.M. as we remove Figure 2 from the main text due to space constraint.

B. Is this for surprisal? Indicate what these figures show. Also, there are no units on the color bar.

Yes, all effects reported in Figure 2 are for surprisal. The figures showed that 1) surprisal elicited a more negative ERP ~300-600ms across most of the electrodes, 2) this surprisal effect is primarily central parietal and 3) Cz and Pz showed N400 like waveform. The color bar represents F values of the electrode, thus there are no units. The information is now added to the caption. See changes in S.M.

C. "red indicates the confidence interval." should be shaded pink because red is taken for the time window. "The red line underlying the figures" should be underlying the graphs.

This is now fixed in the caption. See changes in S.M.

Figure 3

- Consider showing some raw EEG traces also in these figures (from 3 on) to give a better sense of the data.

We cannot show EEG plots in the main text due to page limits, but instead placed it in the S.M.. The EEG plot is also not an accurate illustration of the results, as we dichotomised all continuous variables (e.g. surprisal, pitch prosody and mouth) for illustration.

- From the graph, it seems like for high surprisal words, high F0 reduces the N400, which is discussed in the text, but for low surprisal words, high F0 enhances it, which is not discussed?

This is now addressed on p. 16.

- How do you explain the difference between Exp. 1 and 2? It seems like in Exp 1 the surprisal effect on the N400 is reversed for high F0?

We believe this is mostly driven by the fact that the main effect of surprisal is smaller in Experiment 1 than in Experiment 2. Therefore, when mediated by a positive interaction with prosody, the surprisal effect is reduced but still negative in Exp.2, but becomes positive in Exp.1. The larger surprisal effect in Exp.2 is possibly due to the use of longer passages (45 words on average, compared with 23 in Exp.1).

Discussion:

- “We found that ... each cue ... had a general effect and crucially modulated linguistic-based surprisal.” Again (see p. 2 above), the cues did not modulate the surprisal, but the N400 or **perhaps the effect of surprisal on the N400.**

We fixed this in the discussion.

- “Thus, our results clearly show that language comprehension, in its natural face-to-face ecology, involves more than just speech:” **Because comprehension per se was not measured, but just an EEG measure that is associated with it, it would be good to be more careful in the wording (e.g., ‘language processing’ is more neutral).**

We changed to language processing (as also suggested by reviewer 2).

- “Prosody, gesture and mouth contributes to...” — a typo - contribute rather than contributes

Thank you, we have now fixed.

- “Crucially, meaningful gestures, but not beat gestures, decrease the cognitive load in word processing.” Unclear what is this based on. Talking about ‘cognitive load’ is much too general.

This has been rephrased (p. 16).

- “multimodal ‘non-linguistic’ cues have a central role in processing as they always modulate word predictability”

The authors use the word “always” throughout the paper; it is not clear what ‘always’ is used with respect to—either remove, or clearly specify the alternative.

We have now explained why we use always in the introduction: “we address two key questions about face-to-face multimodal communication. First, to what extent is the processing of multimodal (‘non-linguistic’) cues central to natural language processing? Previous work has shown that these cues are used in certain conditions (e.g., when the linguistic cues are ambiguous (e.g. Holle & Gunter, 2007), when there is incongruency between a multimodal cue and linguistic cues (e.g. Heim & Alter, 2006) often relatively artificial (e.g., face is covered), thus it remains unclear whether listeners always use these cues in naturalistic language processing.”